# TEST-TIME CORRECTION WITH HUMAN FEEDBACK: AN ONLINE 3D DETECTION SYSTEM VIA VISUAL PROMPTING

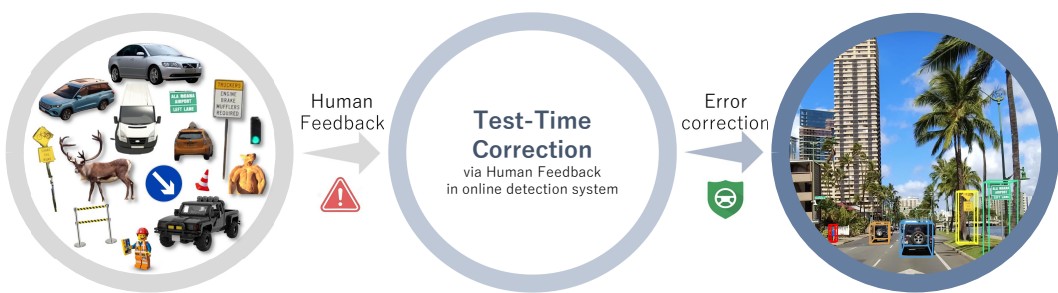

Figure 1: **TTC system** could instantly respond to missed objects via human feedback, detecting and tracking these targets in subsequent frames consistently. As depicted on the left, human feedback might come from any views of target objects, including missed objects in the current and/or previous frames, as well as diverse viewpoints across scenes, styles, and poses. Such a system improves offline-trained 3D detectors by rectifying online driving behavior immediately, reducing safety risks via test-time correction. TTC *enables* a reliable and adaptive online autonomous driving system.

## ABSTRACT

This paper introduces Test-time Correction (TTC) system, a novel online 3D detection system designated for online correction of test-time errors via human feedback, to guarantee the safety of deployed autonomous driving systems. Unlike well studied offline 3D detectors frozen at inference, TTC explores the capability of instant online error rectification. By leveraging user feedback with interactive prompts at a frame, *e.g.*, a simple click or draw of boxes, TTC could immediately update the corresponding detection results for future streaming inputs, even though the model is deployed with fixed parameters. This enables autonomous driving systems to adapt to new scenarios flexibly and decrease deployment risks reliably without additional expensive training. To achieve such TTC system, we equip existing 3D detectors with OA module, an online adapter with prompt-driven design for online correction. At the core of OA module are *visual prompts*, images of missed object-of-interest for guiding the corresponding detection and subsequent tracking. Those visual prompts, belonging to missed objects through online inference, are maintained by the visual prompt buffer for continuous error correction in subsequent frames. By doing so, TTC consistently detects online missed objects and immediately lowers down driving risks. It achieves reliable, versatile, and adaptive driving autonomy. Extensive experiments demonstrate significant gain on instant error rectification over pre-trained 3D detectors, even in challenging scenarios with limited labels, zero-shot detection, and adverse conditions. We hope this work would inspire the community to investigate online rectification systems for autonomous driving post-deployment. Code would be publicly shared.

## 1 INTRODUCTION

Visual-based 3D object detection, which localizes and classifies 3D objects from visual imagery, plays a crucial role in autonomous driving systems. Visual autonomous driving frameworks (Hu et al., 2023; 2022b; Casas et al., 2021; Cui et al., 2021) rely heavily on accurate 3D detection outcomes to predict future driving behaviors and plan the trajectory of the ego vehicle. Existing 3D object detectors (Li et al., 2022; Yang et al., 2023a; Liu et al., 2022; Reading et al., 2021; Huang et al., 2021) typically follow an offline training and deployment pipeline. Once the model is trained and deployed on self-driving cars, it is expensive to update new behaviors, *e.g.*, another turn of offline re-training or fine-tuning. That

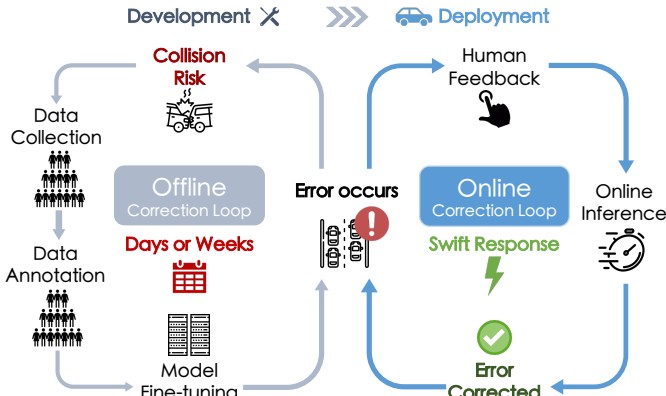

Figure 2: **Comparison of Error Correction** between the conventional offline loop (left) and the new proposed online TTC System (right). Offline error correction pipeline improves model capability during the development stage, which typically requires expensive workloads and computational overhead over days or weeks for model updates. While TTC system additionally enables deployed 3D detectors with on-the-fly error rectification ability.

is, when the system fails to perceive important objects or fails in novel scenarios due to a domain shift, these offline solutions cannot update themselves online to rectify mistakes immediately and detect missed objects. Such a caveat poses significant safety risks to reliable driving systems, *e.g.*, dangerous driving behaviors such as improper lane changing, turning, or even collisions.

To guarantee safety, we argue that 3D detectors deployed on autonomous driving systems can rectify missed detection on the fly during test time. As depicted in Figure 2 (left), for error rectification, existing 3D detectors rely on offline pipelines, encompassing a full-suite procedure of data collection, annotation, training, and deployment. This requires significant human workloads and resources for labeling and re-training, days or even weeks to fulfill. Other than the offline pipeline to improve models, we desire deployed 3D detectors also capable of test-time correction since such delays in offline updating are unacceptable when facing risks on the road, where safety is of the utmost priority.

In this work, we explore a new 3D detection system capable of **T**est-**T**ime error **C**orrection based on human feedback online, namely **TTC**, akin to how human drivers respond, as shown in Figure 2 (right). It is designed to enable existing 3D detectors with immediate error correction ability through online human warnings. Inspired by the principles of In-context Learning Customization (Wei et al., 2022; Peng et al., 2023) in large language models (LLMs) (OpenAI, 2023; Team, 2023), we achieve the TTC system by leveraging images of missed objects collected from human feedback as context prompts. Without extra training, these prompts assist deployed 3D detectors in identifying and localizing previously unrecognized objects in later streaming input frames.

The proposed TTC includes two components: Online Adapter (OA) that enables 3D detectors with visual promotable ability, and a visual prompt buffer that records missing objects. The core design is "visual prompts", the visual object representation derived from human feedback. Existing promptable 3D detection methods typically utilize text, boxes, or clicks as prompts. However, text prompts can be ambiguous and may not describe the target objects effectively. Meanwhile, box and point prompts struggle to handle streaming data. These limitations indicate that such prompts are inadequate for real-time autonomous driving tasks. Visual prompts cover arbitrary imagery views of target objects, *i.e.*, views in different zones, styles, and timestamps (Figure 3), indicating the identity of target objects. With visual prompts, OA module generates corresponding queries, locates corresponding objects within streaming inputs, and facilitates 3D detectors to output 3D boxes.

To enable consecutive error rectification for video streaming, we design a dynamic visual prompt buffer to maintain visual prompts of all past unrecognized objects. In each iteration, we use all

visual prompts in this buffer as inputs of the TTC system. This enables the continuous detection and tracking of all previously missed objects, redressing online errors for streaming input effectively. Further, to refrain the buffer from undesirable expansion, we introduce a "dequeue" operation to ensure its bounded size, allowing for consecutive rectification without excessive overhead.

We conduct experiments on the nuScenes dataset (Caesar et al., 2020). Given the novel setting of our TTC system, the assessment focuses on its abilities for instant online error correction. To this end, we design extensive experiments to verify the key aspects of the overall system: the effects of TTC system in rectifying errors over the online video stream; and the effects of TTC when encountering challenging, extreme scenarios with large amounts of missed detections.

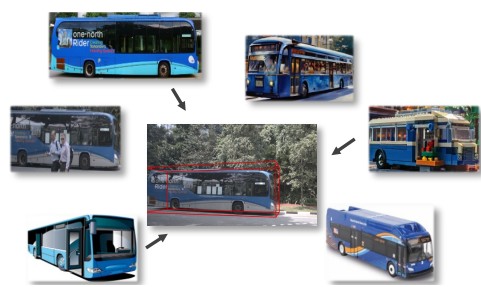

Figure 3: **Visual prompts** could be arbitrary views of objects, across zones, styles, timestamps, *etc.*

Remarkably, the TTC system significantly improves the offline 3D detectors for test-time performance. Specifically, with test-time rectification, TTC system improves offline monocular (Zhang et al., 2022a), multi-view (Wang et al., 2023c), and BEV detectors (Yang et al., 2023a) by 5.0%, 12.7%, and 12.1% EDS [1] , respectively, without requiring any training. Second, when evaluated on challenging scenarios, the TTC system exhibits even more substantial gains, with improvements of 14.4%, 21.6%, 13.6%, and 4.7% EDS on tasks like distant 3D detection and vehicle-focused detection with limited annotations, zero-shot extensions, as well as scenarios with domain shifts, respectively.

This comprehensive evaluation highlights the versatility and adaptability of TTC, which can effectively rectify online errors and maintain robust 3D detection capabilities even under limited data, category shifts, and environmental changes. We hope the introduction of TTC system will inspire the research community to further explore the online rectification approach in autonomous driving systems, a crucial technology that can enhance the safety and reliability of safety-critical applications.

## 2 RELATED WORK

**Interactive Vision Models.** Interactive vision models are designed for tasks based on user inputs. As one of the fundamental tasks in computer vision, they are extensively researched with numerous breakthroughs (Li et al., 2004; Chen et al., 2022; Liu et al., 2023c; Xu et al., 2016; Grady, 2006). Particularly, the advent of the Segment Anything Model (SAM) (Kirillov et al., 2023) has sparked a surge of progress, with applications spanning reconstruction (Shen et al., 2023), detection (Ren et al., 2024; Yang et al., 2023b), segmentation (Zou et al., 2023), image editing (Gao et al., 2023), and more. Compared to existing models, which typically rely on prompts such as clicks, boxes, or scribbles, in this work, we study visual prompts, a new prompt referring to the actual images of objects with arbitrary poses and styles. Visual prompts enable continuous detection and tracking of target objects, facilitating the immediate correction of failed detections at test time. This represents a departure from traditional prompt types, offering more natural and dynamic interactions with the visual content.

**In-context Learning in LLMs.** In-context learning (ICL) (Zhang et al., 2023; Alayrac et al., 2022; Chen et al., 2019), popularized by GPT (Brown et al., 2020), enables LLM for customized interactions with humans. This is then studied for chain-of-thought (Wei et al., 2023) to ensure an informative dialogue system (OpenAI, 2023; Team, 2023; Taori et al., 2023; Chiang et al., 2023; Liu et al., 2023b;a). Existing research demonstrates ICL as an effective mechanism for test-time response, serving as an efficient zero-shot output learner. Inspired by the principles of ICL, we introduce a novel concept called visual prompts, which are zero-shot learners for online 3D detection rectification.

**Online 3D Detection System.** In this paper, we introduce online 3D detection, a new task of instant error rectification for the online testing phase of 3D detectors. The key goal is to enable the continuous detection of objects missed by the offline-trained 3D detectors without additional training. This relates to several areas, such as 3D detection, tracking, continual learning, and open-world active

---

[1] EDS is a class-agnostic version of nuScenes Detection Score (NDS) (Caesar et al., 2020), treating all objects as class-agnostic entities and ignoring the velocity and attribute to evaluate out-of-distribution 3D detection.

Figure 4: **Overall Framework.** (Left:) The TTC system centers on a TTC-3D Detector which utilizes visual prompts $\mathcal{P}_v$ from the visual prompt buffer for test-time error rectification. (Right:) The TTC-3D Detector can be based on any traditional detector (BEV or monocular). It supports 3D detection from any combination of four prompts, *i.e.*, object $\mathcal{P}_o$, box $\mathcal{P}_b$, point $\mathcal{P}_p$, and novel visual prompts $\mathcal{P}_v$, arbitrary views of target objects across scenarios and timestamps.

learning. In contrast to traditional 3D detectors that focus on the performance of trained models (Shi et al., 2020; Lang et al., 2019; Yin et al., 2021; Philion & Fidler, 2020; Li et al., 2022), online 3D detection places greater emphasis on correcting errors during testing. Compared to tracking (Liang et al., 2020; Pang et al., 2021; Park et al., 2008; Zhang et al., 2022b; Hu et al., 2022a), TTC system can track objects across scenes and zones via visual prompts. Different from continual learning (Singh et al., 2021; Ghosh, 2021; Wang et al., 2023a) or active learning for 3D detection (Luo et al., 2023; Chen et al., 2023; Yuan et al., 2023), this task prioritizes the "timeliness" of error rectification, aiming to enhance the test-time performance instantly without any model update or training.

## 3 TEST-TIME CORRECTION WITH HUMAN FEEDBACK

In this section, we elaborate on our TTC, an online test-time error rectification system for 3D detection to detect and track previously missed objects during on-road inference with the guidance of human feedback. We start with an overview of TTC system in Section 3.1, then delve into OA module and the visual prompt buffer in Section 3.2, and Section 3.3, respectively.

### 3.1 OVERVIEW

We convert online human feedback, *i.e.*, clicks, boxes, or uploaded images, into a uniform representation called "visual prompts", which are image descriptions of target objects. Such image-based descriptions can cover arbitrary views of objects, including pictures taken from diverse zones, weather, timestamps, or even from out-of-domain sources such as stylized Internet images. Upon visual prompts, the TTC system, a recurrent framework, is designed to engage in sustained interaction with human users, continuously learning to detect and track new objects. As Figure 4 shows, it comprises two key components: 1) TTC-3D Detector, any 3D detector equipped with OA module for in-context 3D detection and tracking via visual prompts, and 2) an extendable visual prompt buffer storing visual prompts of all previously missed objects, enabling continuous online error rectification.

During online inference after being deployed on cars, whenever an error occurs, *i.e.*, miss an object, users can add the unrecognized object to visual prompt buffer by clicking on it in the image. The model then detects the corresponding 2D and 3D boxes based on the user-provided clicking prompt and updates the visual prompt buffer with the associated image patch. In subsequent frames, TTC-3D Detector leverages the stored visual prompts to detect and track previously missed 3D objects. This enables instant error correction and continuous improvement of 3D detection during online operation.

### 3.2 ONLINE ADAPTER (OA)

OA module is conceived as a bridge between prompts and offline-trained 3D detectors. It receives human prompts and transforms them into queries that can be seamlessly used in traditional detectors. It is flexible to handle four prompts in different forms: object query prompts for traditional offline 3D

detection, box and point prompts for collecting test-time feedback, and visual prompts for consistently correcting errors with streaming video. Specifically, these prompts are processed as follows:

- For object prompts $\mathcal{P}_o$, the OA module generates a set of learnable embeddings as queries, akin to traditional 3D detectors, which are updated during training as demonstrated in previous works (Li et al., 2022; Wang et al., 2021; Zhang et al., 2022a).

- For box $\mathcal{P}_b$ and point $\mathcal{P}_p$ prompts, the OA module encodes them with their location and shape, representing them as Fourier features (Tancik et al., 2020).

- For visual prompts $\mathcal{P}_v$, the OA module first extracts their visual features $\mathcal{Z}_v$ by an encoder (He et al., 2016), then localizes their corresponding objects within the input images $\mathcal{X}_v$, and finally adds their features with the Fourier positional encoding as subsequent inputs:

$$\mathcal{P}_v = \texttt{FourierPE}(\texttt{Align}(\mathcal{Z}_v, \mathcal{Z})) + \mathcal{Z}_v, \tag{1}$$

where $\mathcal{Z}$ means image features of image input, $\texttt{FourierPE}(\cdot)$ is the Fourier positional encoding (Tancik et al., 2020) and the $\texttt{Align}(\cdot)$ operation means "Visual Prompt Alignment", the process of inferring the 2D position in the current frame of the visual prompt.

**Visual Prompt Alignment.** We perform the $\texttt{Align}$ operation to localize target objects in the input images, by visual prompts. To handle flexible visual prompts, which can be image descriptions of objects in any view, scene, style, or timestamp, we employ contrastive mechanisms (Wu et al., 2018; He et al., 2020) as the key design to retrieve target objects at different styles. This allows the module to detect the target objects effectively, even when they exhibit diverse visual styles and appearances.

Specifically, we use two multi-layer perceptrons (MLPs) to first align the channels of image features $\mathcal{Z}$ and visual prompt features $\mathcal{Z}_v$. We then compute the dot product between the aligned feature maps to obtain a similarity map. To further retrieve the coordinates of target objects from the similarity map, we multiply it with the original image features $\mathcal{Z}$, and finally apply another MLP with two output channels to regress the spatial positions $\mathcal{X}_v$ of target objects.

**Instance Ambiguity & Loss.** Sometimes, visual prompts might exhibit instance ambiguity, where multiple objects in the image match the visual descriptions of prompts. For example, suppose the visual prompt describes a traffic cone and several similar-looking traffic cones present in the input images. It can be challenging to uniquely identify the specific object-of-interest (See Fig. 9).

For such cases, we design to retrieve all objects with similar identities to the visual prompt. Specifically, we modify the align operation to predict multiple spatial coordinates $\mathcal{X}_v = \{\mathcal{X}_v^{(i)}\}, i \in \{1, 2, ..., N\}$ for each visual prompt, and add the Fourier features of those $N$ coordinates to the visual prompt features to indicate visual prompts in different positions. $N$ is set to 4 in our implementation.

For similarity supervision, we generate binary segmentation labels based on the ground-truth 2D boxes. Focal loss (Lin et al., 2017) and Dice loss (Milletari et al., 2016) are used for optimization. To supervise the visual prompt localization $\mathcal{X}_v$, we use the Smooth-$l1$ loss with the target as the center coordinates of ground-truth 2D bounding boxes. For dealing with instance ambiguity, we refer to SAM (Kirillov et al., 2023) and only backpropagate the sample with the minimum localization loss during each training iteration. For more details, please refer to the appendix.

**Model Design.** The overall mechanism of TTC-3D Detector is depicted in Figure 4 (right). Based on any traditional offline-trained 3D detector, BEV detector, or monocular detector, we integrate OA module and train it to be promptable. Specifically, OA module takes features extracted by the image encoder of the corresponding 3D detector, along with various forms of prompts as inputs. It encodes these prompts and generates a series of queries, represented as $\mathcal{P} = \{\mathcal{P}_o; <\mathcal{P}_b, \mathcal{P}_p, \mathcal{P}_v>\}$, where $< ... >$ denotes an arbitrary combination of different prompts. These queries are then fed into the transformer decoder of the 3D detector to output 3D boxes following human feedback.

### 3.3 VISUAL PROMPT BUFFER

Visual prompt buffer is a queue that stores user-provided visual prompts of missed objects during online inference. The flexibility of the TTC system lies in allowing users to freely select and define visual prompts, which can be either image contents from the current scene or customized objects from the Internet. This versatility makes the TTC system applicable to a wide spectrum of scenarios.

**Dequeue.** To prevent the buffer from growing indefinitely, we design a "dequeue" mechanism. We filter out visual prompts with low confidence, as they likely no longer appear in the scene. We also use the intersection-over-union (IoU) between predictions to identify and remove redundant visual prompts. This dynamic update and maintenance is the key of TTC system, ensuring a balance between latency and accuracy by adaptively incorporating and pruning feedback during online inference.

## 4 EXPERIMENTS

We now proceed to evaluate our TTC system. Compared to traditional 3D detection approaches, the evaluation of TTC system should consider more on its capabilities in instant online error correction over offline-trained 3D detectors. Thus, we assess the system from two key aspects:

- How does the TTC system perform with various adapted offline 3D detectors for instant online error rectification?
- How does the TTC system enhance offline-trained 3D detectors when faced with out-of-training-distribution scenarios?

### 4.1 EXPERIMENTAL SETUP

**Dataset.** Experiments are done on nuScenes dataset (Caesar et al., 2020) with 1,000 autonomous driving sequences, one of the most popular datasets for autonomous driving research.

**Tasks.** Due to the new setting, experiments are conducted in two aspects to answer the questions above. The first experiment aims to test the TTC system in boosting the performance of offline-trained 3D detectors without re-training via online error rectification. We conduct this verification by applying TTC system to various established offline 3D detectors (Zhang et al., 2022a; Li et al., 2022; Yang et al., 2023a; Liu et al., 2024; Wang et al., 2023b; Lin et al., 2023; Wang et al., 2023c) We mimic user intervention during inference. Specifically, we simulate the user's online missing feedback by comparing the distance between ground truth and detected 3D boxes. Ground-truths without any detection within 2m are considered missed and added into the prompt buffer.

Then, we validate the TTC system to correct detection errors in out-of-training-distribution scenarios. To conduct a thorough quantitative analysis of this aspect, we established four tasks under different settings: **(a.)** discarding 80% 3D bounding box labels of distant objects farther than 30m during training, and test the error corrections for detecting distant objects; **(b.)** discarding 80% 3D bounding box labels of instances labeled as vehicle, including "car", "truck", "C.V.", "bus", and "trailer", and test the error correction in vehicle objects; **(c.)** discarding all 3D bounding box annotations of class "truck" and "bus", and test the zero-shot ability of TTC on those discarded classes; **(d.)** discarding all training data of the scenario of "Nighttime" and "Rainy", and test the improvements of TTC on scenarios with domain gap. For these tasks, we base TTC system on the monocular algorithm, MonoDETR (Zhang et al., 2022a), and test it under each set separately, as the monocular setting demonstrates the most general applicability. Based on TTC-MonoDETR, we also present qualitative results to show the potential of our system to address corner cases in challenging real-world scenarios through human feedback during test time. Through these experiments, we demonstrate TTC as an effective system to adapt offline 3D detectors to challenging scenarios, without training.

**Entity Detection Score.** As we focus on enabling existing 3D detectors to rectify missing detections instantly in an online manner, we set our priority on the model's ability to zero-shot localize and detect new objects during the testing phase, rather than its classification capability. Therefore, we remove all class annotations of 3D objects during training and simply treat all objects as entities (Kirillov et al., 2023; Qi et al., 2022). For evaluation, we use the Entity Detection Score (EDS), the class-agnostic version of the nuScenes Detection Score (NDS), to assess the performance, which emphasizes the localization quality of target objects. More details can be found in Appendix B.3.

**Implementation Details.** We implement our method based on mmDet3D codebase (Contributors, 2020), and conduct all experiments on a server with $8\times$ A100 GPUs. In OA module, we use a ResNet18 (He et al., 2016) to extract visual prompt features. All visual prompts are resized to $224 \times 224$ before being sent to OA module. For training, we use AdamW (Kingma & Ba, 2015; Loshchilov & Hutter, 2019) optimizer with a batch size of 16, equally distributed to 8 GPUs. We

Table 1: **Effect of TTC system on various 3D detectors.** TTC system effectively improves the test-time performance of offline-trained detectors during online inference without any extra training.

| Method | Backbone | Type | mAP (%) ↑ | EDS (%) ↑ |
|---|---|---|---|---|
| MonoDETR (Zhang et al., 2022a) | R101 | Monocular | 37.9 | 38.2 |
| TTC-MonoDETR | | | 42.6 (**+5.7**) | 43.2 (**+5.0**) |
| MV2D (Wang et al., 2023c) | R50 | Multiview | 38.9 | 38.3 |
| TTC-MV2D | | | 51.0 (**+12.1**) | 51.0 (**+12.7**) |
| Sparse4Dv2 (Lin et al., 2023) | R50 | Multiview + Temporal | 38.8 | 37.6 |
| TTC-Sparse4Dv2 | | | 53.5 (**+14.7**) | 52.1 (**+14.5**) |
| BEVFormer (Li et al., 2022) | R101 | BEV + Temporal | 36.5 | 35.8 |
| TTC-BEVFormer | | | 47.8 (**+11.3**) | 47.2 (**+11.4**) |
| BEVFormerV2-t8 (Yang et al., 2023a) | R50 | BEV +Temporal | 39.6 | 38.9 |
| TTC-BEVFormerV2-t8 | | | 51.6 (**+12.0**) | 51.0 (**+12.1**) |
| RayDN (Liu et al., 2024) | R50 | BEV + Temporal | 39.6 | 38.7 |
| TTC-RayDN | | | 51.7 (**+12.1**) | 50.8 (**+12.1**) |
| StreamPETR (Wang et al., 2023b) | V2-99 | BEV + Temporal | 39.7 | 39.1 |
| TTC-StreamPETR | | | 52.3 (**+12.6**) | 51.5 (**+12.4**) |

Table 2: **Experiments on out-of-training-distribution scenarios.** TTC system achieves substantial gains with limited or even no labels under different challenging test cases.

(a) **Long-range rectification.** Effect of TTC system in detecting distant objects with 20% annotations.

| Model | All (0m-Inf) | | Dist. (30m-Inf) | |
| Setting | mAP (%) | EDS (%) | mAP (%) | EDS (%) |
|---|---|---|---|---|
| Point | 41.6 | 40.8 | 17.8 | 19.4 |
| Box | 44.3 | 43.7 | 19.2 | 21.7 |
| Visual | 42.6 | 42.0 | 18.3 | 21.6 |
| MonoDETR | 31.4 | 31.6 | 0.0 | 0.0 |
| TTC-MonoDETR | **40.2** | **40.1** | **11.0** | **14.4** |
| Δ | +8.8 | +8.5 | +11.0 | +14.4 |

(b) **Vehicle-focused rectification.** Effect of TTC system on vehicle objects with 20% annotations.

| Model | All | | Vehicle | |
| Setting | mAP (%) | EDS (%) | mAP (%) | EDS (%) |
|---|---|---|---|---|
| Point | 38.0 | 36.9 | 33.5 | 36.6 |
| Box | 41.2 | 40.0 | 36.7 | 39.6 |
| Visual | 39.2 | 38.4 | 34.6 | 38.5 |
| MonoDETR | 17.6 | 16.1 | 2.4 | 7.3 |
| TTC-MonoDETR | **29.0** | **27.2** | **23.2** | **28.9** |
| Δ | +11.4 | +11.1 | +20.8 | +21.6 |

(c) **Novel object rectification.** Effect of TTC system on objects of novel classes unseen in the training set.

| Model | All | | Unseen | |
| Setting | mAP (%) | EDS (%) | mAP (%) | EDS (%) |
|---|---|---|---|---|
| Point | 35.2 | 35.6 | 11.9 | 15.0 |
| Box | 38.5 | 38.8 | 14.7 | 16.9 |
| Visual | 35.4 | 36.0 | 11.2 | 15.6 |
| MonoDETR | 28.3 | 29.6 | 0.0 | 0.0 |
| TTC-MonoDETR | **34.8** | **36.1** | **8.5** | **13.6** |
| Δ | +6.5 | +6.5 | +8.5 | +13.6 |

(d) **Domain shift rectification.** Effect of TTC system on objects in scenarios with domain gap.

| Model | All | | Rain & Night | |
| Setting | mAP (%) | EDS (%) | mAP (%) | EDS (%) |
|---|---|---|---|---|
| Point | 39.0 | 38.2 | 30.5 | 30.7 |
| Box | 42.6 | 41.5 | 33.4 | 33.6 |
| Visual | 39.8 | 39.4 | 29.4 | 30.8 |
| MonoDETR | 34.5 | 34.7 | 25.2 | 26.6 |
| TTC-MonoDETR | **39.9** | **40.0** | **29.7** | **31.3** |
| Δ | +5.4 | +5.3 | +4.5 | +4.7 |

initialize the learning rate as 2e-4, adjusted by the cosine annealing policy. When training point and box prompts, we simulate user inputs with noise by adding perturbations to the ground truth. To ensure the visual prompts are robust across multiple scenes, timestamps, and styles, we choose visual prompts of target objects not only from the image patches of the current frame but also randomly from previous and future frames within a range of $\pm 5$ when training. Flip operations are used as data augmentations. During testing, we remove redundant predictions by non-max-suppression (NMS) with IoU and classification confidence thresholds as 0.5 and 0.3, respectively.

## 4.2 MAIN RESULT

**Effectiveness of TTC System in Test-time Error Correction.** Test-time error correction ability without re-training is the core capability of TTC system. We verify this by incorporating traditional offline-trained 3D detectors into TTC system and compare the performance without re-training. For thorough verification, we select various offline-trained 3D detectors, including monocular (Zhang et al., 2022a), multi-view (Wang et al., 2023c; Lin et al., 2023), and BEV ones (Li et al., 2022; Yang et al., 2023a; Wang et al., 2023b; Liu et al., 2024). As shown in Table 1, the TTC system substantially improves the test-time performance of offline-trained 3D detectors, *e.g.*, 11.4% and 12.4% EDS

Figure 5: **Qualitative visualization of real-world scenes (collected from YouTube).** We visualize the zero-shot 3D detection results in a real-world scenario. In this case, the prompt buffer contains a visual prompt of a deer. Higher responses from the visual prompt alignment are highlighted by brighter colors. As shown, although **trained solely on nuScenes**, TTC system can still accurately localize "unseen" objects in the input image. Best viewed in color.

improvements on BEVFormer and StreamPETR, without requiring any training. These demonstrate the effectiveness of the TTC system in instantly correcting test-time errors during online inference.

**Effectiveness of TTC System in Out-of-Training Scenarios.** Table 2 presents results to validate the TTC system in challenging and extreme scenarios. In these experiments, we base TTC system on MonoDETR for its simple monocular setting. "Point", "Box", and "Visual" in Table 2 are TTC-MonoDETR using point, box, or image patch of target objects in the input image as input prompts. These serve as performance upper bounds as they receive human feedback in every frame.

Table 2a evaluates the effectiveness of TTC system in rectifying detection errors on distant objects (beyond 30m). During training, 80% of the far-away annotations are removed, resulting in a 3D detector with poor long-range performance (0.0% mAP and EDS). Powered by the TTC system, the performance on these distant objects is instantly improved to 11.0% mAP and 14.4% EDS, without any training. The experiment is then extended to all vehicles in the nuScenes dataset, as shown in Table 2b. The TTC system achieves impressive performance gains of 20.8% mAP and 21.6% EDS.

Furthermore, in Table 2c and Table 2d, we evaluate the TTC system in two challenging scenarios: encountering unseen objects not present in the training data; handling domain shifts, such as transitioning from sunny to rainy or nighttime conditions. In Table 2c, the TTC system shows effectiveness in successfully detecting novel objects not labeled in the training set, achieving 8.5% mAP and 13.6% EDS on these novel objects, significantly improving the offline-trained baseline with 0.0% mAP and EDS. In Table 2d, we find TTC also works well for online error rectification when driving into scenarios with domain shifts, providing 4.5% mAP and 4.7% EDS improvements.

Regarding qualitative results, Figure 5 further illustrates a case of the zero-shot capability in real-world scenarios. Despite being trained solely on the nuScenes dataset, TTC-MonoDETR can detect a `Deer` using a visual prompt of another deer from a different viewpoint, which is extremely challenging for traditional offline detectors. More qualitative examples can be found in Appendix D.4.

These experiments demonstrate TTC as an effective and versatile system for instant error correction, excelling at handling missing distant objects, unseen object categories, and domain shifts, without requiring any training. The superior performance of TTC system in these challenging real-world scenarios highlights its potential to enable robust and adaptable 3D object detection systems.

## 4.3 ROBUSTNESS OF VISUAL PROMPTS

As a crucial component of the continuous test-time error correction system, we conduct a series of ablation studies on visual prompts in this section. We base TTC system on MonoDETR and validate its robustness concerning prompts obtained from the Internet or those from different moments.

**Robustness over Web-derived Visual Prompts.** Visual prompts can be arbitrary imagery views of target objects and can be from any image source. For example, we can use images sourced from the Internet as visual prompts. We fix the prompt buffer with visual prompts from the Internet during inference, and assess the effectiveness of handling visual prompts with diverse styles. We employ the model from Table 2b (TTC-MonoDETR trained with 20% labels of vehicles), and select 12 car and 6

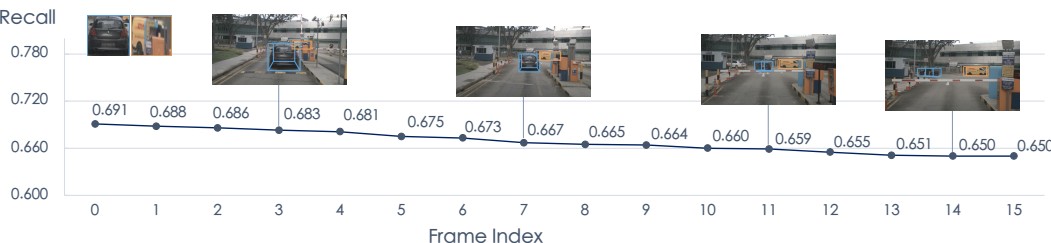

Figure 6: **Visual prompt examples derived from the Internet.** We select 12 cars and 6 buses with different views and colors from websites to cover the distribution of object appearances in the nuScenes dataset comprehensively.

Table 3: **Results with web-derived visual prompts.** TTC-MonoDETR[†] indicates the model with frozen prompt buffer containing pre-assigned visual prompts derived from the Internet. TTC system can still significantly improve traditional 3D detectors even using web prompts in scenarios with very limited labeled data.

| Category | Method | mAP (%) ↑ | EDS (%) ↑ |
|---|---|---|---|
| Car | MonoDETR | 3.2 | 9.1 |
| | TTC-MonoDETR[†] | 20.9 (**+17.7**) | 27.5 (**+18.4**) |
| Bus | MonoDETR | 0.4 | 4.4 |
| | TTC-MonoDETR[†] | 17.4 (**+17.0**) | 20.6 (**+16.2**) |

Figure 7: **Experiments on visual prompts from arbitrary temporal frames.** TTC-3D Detectors can effectively locate and detect target objects in future frames, using its image patch at Frame #0.

bus images from the Internet, which resemble those in nuScenes dataset, as visual prompts for online correction. We show them in Figure 6. As listed in Table 3, TTC system demonstrates strong online correction capabilities though with limited annotations. Even with prompts sourced from the Internet with various styles and poses, TTC system still improves the offline-trained baseline by 18.4% EDS on "Car" objects under this extremely challenging setting. This further underscores the robustness of TTC system and highlights its potential to address the long-tail challenges in real-world scenarios.

**Robustness over Arbitrary Visual Prompt Views.** We also investigate the capability of TTC system in handling visual prompts of target objects across scenes and times. In Figure 7, we study whether our system can successfully associate objects with its visual prompts from arbitrary frames. Specifically, for each video clip of the nuScenes validation set, we use the image patch of target objects in the first frame as visual prompts, then detect and track target objects in subsequent frames, and compute the recall rate for evaluation with arbitrary views. Figure 7 presents the results of TTC-MonoDETR, showing that despite significant differences in viewpoints and object poses between frames, the recall rate does not drop dramatically as the ego vehicle moves. This highlights the robustness of the TTC detectors in handling visual prompts with arbitrary views across scenes and times, indicating that the TTC system can effectively process human feedback that may involve temporal delays.

## 5 CONCLUSION

In this paper, we introduce the TTC system. It equips existing 3D detectors with the ability of test-time error correction. The core component is the OA module, which enables offline-trained 3D detectors with the ability to leverage visual prompts for continuously detecting and tracking previously missing 3D objects. By updating the visual prompt buffer, TTC system enables continuous error rectification online without any training. To conclude, TTC provides a more reliable online 3D perception system, allowing seamless transfer of offline-trained 3D detectors to new autonomous driving deployments. We hope this work will inspire the development of online correction systems.

**Limitations and Future Work.** The current system is limited in scale, including both model and data scale. For future work, we would focus on improving the generalization capabilities of visual promptable online 3D detectors at scale. We plan to combine large-scale 2D detection datasets with limited 3D detection datasets, together with the design of large-scale vision models, to develop a general, versatile, and robust 3D detection system leveraging visual prompts. This proposed research direction aims to advance the state-of-the-art in prompt-based 3D object detection, enabling highly generalizable and reliable systems for automated real-world applications.

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

# Appendix

## A DISCUSSIONS

For better understanding of our work, we supplement intuitive questions that one might raise. Note that the following list does *not* indicate whether the manuscript was submitted to a previous venue.

**Q1:** *What is the relationship between the online 3D detection and the offline data-loop progress?*

We emphasize, in this paper, we do not propose the online system to replace the traditional offline data loop. As illustrated in Figure 2 of the main paper, these two systems address different aspects of autonomous driving. The offline system remains crucial for enhancing the capabilities of the base perception model through development; while our online TTC system further enables the deployed frozen model in vehicles to promptly rectify dangerous driving behaviors caused by unrecognized objects on the road. With improved offline-trained detectors, the TTC can effectively correct more online errors during test-time inference without re-training, as detailed in Table 1.

**Q2:** *What are the main technical novelty and advantages of the proposed TTC system over previous instruction-based 3D detectors?*

The advantages lie in the design of visual prompts, the visual descriptions of target objects with diverse sources, styles, poses, and timestamps. While existing instruction-based 3D detection methods typically utilize text, boxes, or clicks as prompts.

Compared to text prompts, visual prompts provide a more natural and accurate description of the target object. In contrast, verbal descriptions can be ambiguous to convey instance-level features, leading to an inaccurate understanding of the missing objects. Second, text promptable models are often combined with LLMs with high latency and are thus unavailable for autonomous driving deployment, as discussed in Appendix C.6.

Box and point prompts are less convenient than visual prompts when dealing with stream data. If missing occurs, these single-frame prompts require users to provide feedback at every frame, which is unfeasible in real-world applications. Compared to box and point prompts, visual prompts are robust across different scenes and timestamps, one single-frame visual prompt is enough for detecting and tracking in later frames. Furthermore, visual prompts enable 3D detection with pre-defined visual descriptions of target objects, regardless of the sources, styles, poses, etc, as discussed in Section 4.3.

The introduction of novel visual prompts enables real-time, accurate, and continuous error correction of streaming inputs with "one-click" feedback.

**Q3:** *What is the relationship between the TTC system and existing 3D perception tasks?*

The TTC system relates to several 3D perception tasks, including 3D object detection, zero/few-shot detection, domain adaptation, single object tracking, and continual learning.

Compared to standard 3D object detection, the primary focus of the TTC system is on enabling instant online error correction rather than optimizing the offline detection performance of the base 3D detector. In contrast to traditional few-shot, one-shot, or domain adaptation approaches, the TTC system does not require 3D annotations for new objects or any model retraining, yet can still provide reasonable 3D bounding box estimates for out-of-distribution objects. Relative to single object tracking, the TTC system does not rely on bounding boxes of the target objects in the first frame. Instead, it can perform tracking using the visual descriptions of the target objects from any scene or timestamp, leveraging the diverse set of visual prompts.

In summary, the TTC system represents a more flexible and comprehensive 3D object detection framework, combining the strengths of zero-shot detection, handling out-of-distribution objects, and utilizing diverse visual prompts beyond the current scene context.

**Q4:** *Why choose 3D detection as the experimental scenarios of TTC? Could the proposed framework be extended to 2D detection or other vision tasks?*

TTC represents a general idea to equip deployed systems with the capability of online error rectification, making them more versatile, adaptive, and reliable. This idea can be readily extended to other vision tasks, such as 2D detection, for rapid adaptation of pre-trained models to novel scenarios.

The choice of 3D detection for autonomous driving as the experimental scenario is motivated by the paramount importance of safety for deployed self-driving systems. Without an online correction method, mistakes made by the offline model pose significant safety risks for on-road autonomy.

Therefore, we select the autonomous driving domain as the testbed for the TTC framework, given the critical need for a robust, adaptive online error correction system to ensure the reliability of these safety-critical applications. We mark the extension of TTC to other vision tasks as future works.

**Q5:** *What are potential applications and future directions of TTC?*

We believe that, visual prompts, as the core design element of the TTC system, represent a more natural and intuitive query modality for the image domain. This approach has significant research potential and application prospects in the field of 3D perception and beyond.

For example, visual prompts enable rapid customization of the tracking targets, beyond the pre-defined object classes. Second, the visual prompt-based framework facilitates online continual learning for 3D perception systems, adapting to evolving environments. Then, visual prompts can be applied in the V2X domain to enable swift error rectification across diverse operational scenarios. Visual prompts can also be deployed to assist in the auto-labeling process of target objects. Furthermore, by combining visual prompts with natural language prompts, we can obtain more precise descriptions and behavioral control for online perception systems.

The diverse applications outlined above demonstrate the promise of visual prompts as a versatile approach. As showcased in this work, the visual prompt-based framework opens up new possibilities for online perception systems, not only in autonomous driving but also in a broader range of domains.

# B  IMPLEMENTATION DETAILS

## B.1  DETAILS OF TTC SYSTEM

This section elaborates on the detailed workflow of the TTC system. As shown in Figure 8, during online inference, once the TTC-3D Detector fails to recognize an object, users can click on the missing object within the image. Based on the user-provided click, the TTC-3D Detector identifies the corresponding 2D and 3D boxes, crops the relevant areas to obtain visual prompt patches, and subsequently updates the visual prompt buffer. In later frames of inference, the TTC-3D Detector applies these stored visual prompts to continuously detect and track previously missing 3D objects, achieving instant error correction dur-

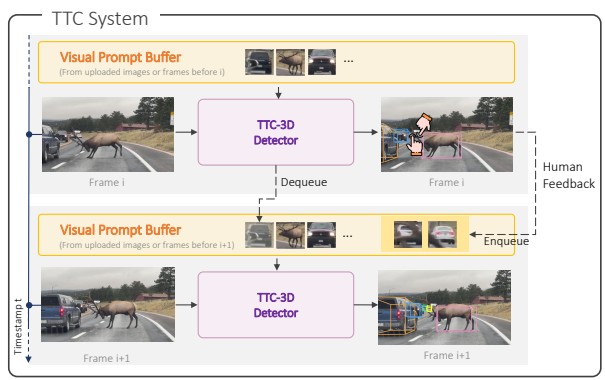

Figure 8: Details of the TTC system.

ing test-time and constantly enhancing the offline-trained 3D detectors after being deployed.

## B.2  VISUAL PROMPT ALIGNMENT

We now delve into the implementation details of visual prompt alignment. As described in Figure 9, given the visual prompt $\mathcal{P}_v$, we first extract the visual prompt features $\mathcal{Z}_v$ using a lightweight encoder, *e.g.*, ResNet18 (He et al., 2016) in our implementation. Then, we use two separate 2-layer perceptrons, each with 128 and 64 output channels, to align the channel dimensions of the prompt features and the input image features. This results in the aligned feature maps $\hat{\mathcal{Z}}_v \in \mathbb{R}^{M \times 64}$ and $\hat{\mathcal{Z}} \in \mathbb{R}^{KHW \times 64}$, where $M$ is the number of visual prompts, $K$ is the number of image views, $H$ and $W$ are the spatial dimensions of image features. Finally, we compute the cosine similarity ($\odot$ in Figure 9) between $\hat{\mathcal{Z}}_v$ and $\hat{\mathcal{Z}}$ to obtain the similarity map $\mathcal{S} \in \mathbb{R}^{M \times KHW}$, which encodes the alignments between the visual prompts and image features.

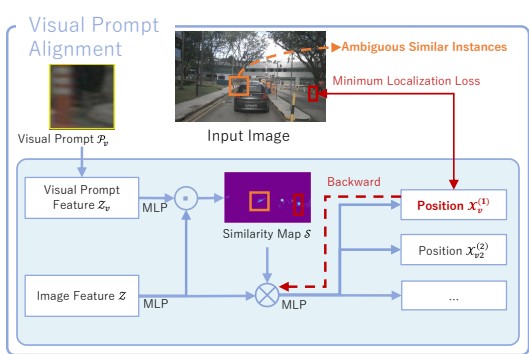

Figure 9: Concrete mechanism of visual prompt alignment. This figure illustrates monocular input. When multi-view images are employed, this alignment operation flattens the different views and still generates $N$ peak candidate positions.

To solve the instance ambiguity issue, we propose to predict multiple spatial coordinates of different peak responses in the similarity map (highlighted by "**orange box**" in Figure 9). Specifically, we first multiply the image features $\mathcal{Z}$ with similarity map $\mathcal{S}$ ($\bigotimes$ in Figure 9), and then use a 2-layer perceptron with $N \times 2$ output channels to regress spatial coordinates of the $N$ peak responses. During training, we only backpropagate the localization loss for the positions that have the minimum loss with respect to the ground truth. In our implementation, $N$ equals 4. This multi-instance retrieval approach allows the TTC to handle cases where the visual prompt matches multiple candidate objects in the input image, improving the robustness of the online error correction.

### B.3 DETAILS OF ENTITY DETECTION SCORE (EDS)

EDS is a class-agnostic version of nuScenes Detection Score (NDS), but further prioritizes the localization quality of target objects. Inspired by (Yang et al., 2024), we improve the original NDS computation by multiplying the recall rate and the mean True Positive metrics, $\mathbb{TP}$, as well, as illustrated below:

Table 4: Feasibility Analysis of the EDS Metric.

| Method | mAP (%) | w.o. Rec (%) | + Rec (%) |
|---|---|---|---|
| BEVFormer | 36.5 | 49.1 | 35.8 |
| TTC-BEVFormer | 47.8 | 54.6 | 47.2 |
| BEVFormer-V2 | 39.6 | 49.2 | 38.9 |
| TTC-BEVFormer-V2 | 51.6 | 58.5 | 51.0 |

$$\text{EDS} = \frac{1}{6}[3\text{mAP} + \text{Recall} \times \sum_{\text{mTP} \in \mathbb{TP}} (1 - \min(1, \text{mTP}))], \tag{2}$$

The intuition behind this is simple. The larger the recall rate is, the more predictions are involved in the statistics of mTP. Compared to simply setting a recall threshold (Caesar et al., 2020), the multiplication adjusts the weight of mTP to EDS according to its comprehensiveness and thus brings a more informative quantitative result. In Table 4, we analyze the effectiveness of this metric. The comparisons between multiplying recall rate or not on various TTC-3D Detectors show that EDS, incorporating recall into its calculation, does not alter the original overall trend. Furthermore, it effectively highlights the superiority of detecting missed objects, demonstrating its validity.

## C ABLATION STUDIES

In this section, we conduct a series of ablation studies. We base our TTC system on MonoDETR for its simple and efficient monocular setting except for experiments in Appendix C.7.

### C.1 EFFECT OF COMPONENTS IN VISUAL PROMPT ALIGNMENT.

Visual prompt alignment aims to localize objects via visual prompts in input images. We now evaluate its components. Table 5 presents experiments to verify the core components of this alignment, including similarity loss (Focal and Dice loss) and localization loss (supervising visual prompt localization, $\mathcal{X}_v$). While the alignment can be implicitly learned by attention mechanisms in the transformer decoder, incorporating explicit similarity supervision brings improvements of 6.8% mAP and 6.6%

Table 5: Effect of visual prompt alignment.

| Sim. Loss | Loc. Loss | mAP (%) | EDS (%) |
|---|---|---|---|
| - | - | 32.6 | 31.9 |
| $\checkmark$ | - | 39.4 | 38.5 |
| $\checkmark$ | $\checkmark$ | **43.3** | **42.8** |

EDS. Further utilizing the position loss and one-to-$N$ mapping (to address instance ambiguity) boosts the performance to 43.3% mAP and 42.8% EDS. These results prove this alignment operation is a critical component enabling the TTC detectors to effectively detect target objects via visual prompts.

### C.2 INSTANCE AMBIGUITY IN VISUAL PROMPT ALIGNMENT

To solve the instance ambiguity issue, we propose to regress $N$ positions of each visual prompt when performing the visual prompt alignment. This retrieves all objects with similar visual contents. In this study, we validate the effectiveness of this design by conducting ablation studies on the number of $N$. As listed in Table 6, when the number of position prediction $N$ equals 1, which means a one-to-one mapping for each visual prompt, the mAP and recall rate

Table 6: Effect of the number of predicted positions $N$ of visual prompt alignment.

| No. of Position Predictions | mAP (%) ↑ | Recall (%) ↑ |
|---|---|---|
| 1 | 39.9 | 62.1 |
| 4 | **43.3** | 69.1 |
| 8 | 43.0 | **69.4** |

are 39.9% and 62.1%, respectively. Then, if we increase the $N$ to 4, effectively a one-to-four mapping, we obtain an mAP of 43.3% and a recall rate of 69.1%. This represents a 7% improvement in the recall rate, demonstrating that instance ambiguity is an important challenge in visual prompt-based detection, and the proposed one-to-$N$ mapping solution effectively addresses this issue.

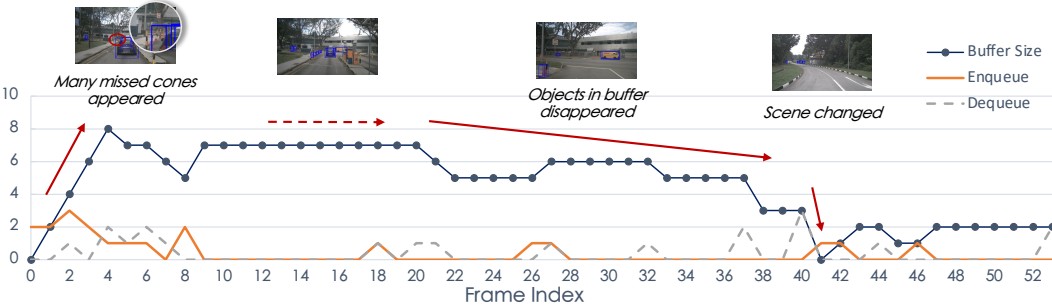

Figure 10: **Size of visual prompt buffer during the video stream.** Visual prompt buffer stores the missed objects during online inference to rectify test-time errors of deployed 3D detectors. It can adaptively manage the stored prompts and thus maintain the balance between latency and accuracy.

## C.3 EFFECT OF VISUAL PROMPT ALIGNMENT ON INSTANCE AWARENESS

a.
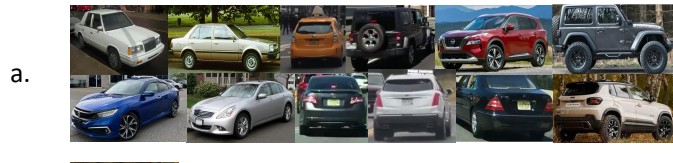

b.
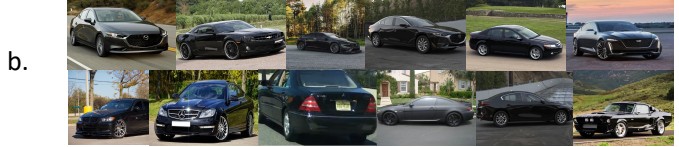

Table 7: **Effect of visual prompt alignment on instance aware- ness.** Group (a.) resembles "car" objects in nuScenes validation set with diverse types and colors; while group (b.) maintains black sedans only. The performance gap demonstrates that alignment learns the instance awareness.

| Group | mAP (%) ↑ | EDS (%) ↑ |
|-------|-----------|-----------|
| a. | 20.9 | 27.5 |
| b. | 13.8 | 18.9 |

Figure 11: Visual prompt showcase for Table 7.

Despite demonstrating that visual prompt alignment can enhance system performance, we remain uncertain whether the alignment can distinguish different objects based on visual prompts for instance-level matching. To investigate, we conduct an additional experiment, based on a similar setting with Table 3[2], but fix the visual prompt buffer with images of black sedans solely (Figure 11). As shown in Table 7, as "Car" objects in nuScenes contain various types and colors, solely using black sedans as visual prompts leads to an 8.6% EDS drop. This underscores that the alignment operation effectively differentiates objects based on visual prompts, achieving instance-level matching and detection.

## C.4 STATISTICS OF VISUAL PROMPT BUFFER DURING ONLINE INFERENCE.

We design the visual prompt buffer to store missed objects during inference with video stream and introduce a "dequeue" mechanism to prevent the buffer from growing indefinitely. In this ablation study, we analyze the dynamic buffer size, as well as the number of enqueued and dequeued in each frame, to illustrate the behavior of visual prompt buffer during the online operation of TTC.

As shown in Figure 10, the visual prompt buffer exhibits three distinct behaviors during online inference in each nuScenes video clip: increasing, steady, and decreasing. In initial frames, many traffic cones are queued into the buffer due to the poor performance of deployed offline 3D detectors on cone objects. The buffer size thus grows quickly in initial frames to store visual prompts of missed objects for online rectification (Frames #0 to #4). The buffer size then stabilizes as the online detector consistently detects and tracks all objects of interest (Frames #4 to #20). Further, as the ego vehicle drives out of the scene, many previously enqueued objects no longer exist and are thus removed from the buffer automatically (Frames #20 to #40). The buffer finally becomes empty as the scene changes.

---

[2]For reference, we employ the TTC-MonoDETR trained on 20% vehicle annotations and freeze the prompt buffer with predefined web prompts when inference.

This demonstrates the effectiveness of visual prompt buffer, which consistently stores missed objects during online inference and corrects online errors. This dynamic behavior, exhibiting increasing, steady, and decreasing phases, highlights itself to manage stored visual prompts for robust 3D object detection performance throughout the online inference with the balance between latency and accuracy.

## C.5 VISUAL PROMPTS WITH USER PERTUBATIONS

In Section 4.3, we have demonstrated the robustness of our system on visual prompts from diverse sources, styles, poses, scenes, and timestamps. Considering visual prompts during online inference are derived from user clicking, this will have positional deviations to the perfect 2D center of the target object. Thus, we design experiments with positional perturbations during test-time inference to analyze the impact. Expressly, we set the maximum translation ratios at 0%, 10%, 20%, 30%, and 40% to the ground-truth 2D centers, and input them to TTC system

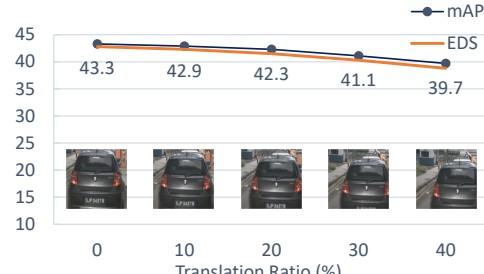

Figure 12: Robustness against feedback noise.

to evaluate the robustness against such disturbances. Figure 12 shows that the TTC-MonoDETR exhibits minor performance drops under increasing positional perturbations, demonstrating strong robustness to these disturbances and highlighting significant potential for real-world applications.

## C.6 ANALYSIS OF MODEL SIZE AND LATENCY

We further analyze the model size and latency of TTC system, primarily comparing them with recent LLM-based promptable 3D detection methods (Chen et al., 2024; Huang et al., 2023). We argue that, for online promptable systems that are developed for real-world applications, latency can come from into two parts. The first one is the unavoidable delay caused by humans

Table 8: Parameters and latency comparisons between LLM based 3D detectors, traditional 3D detectors, and related TTC 3D detectors.

| Method | LL3DA | MonoDETR | TTC-MonoDETR |
|---|---|---|---|
| #Params | 118M | 68M | 80M |
| FPS (Hz) | 0.42 | 11.1 | 9.1 |

from observing the error to reacting to provide prompts. This is inherent to any online prompt-based approach. The other one is the delay associated with the inference speed of the system itself. As the first one cannot be controlled by the system design, we focus on the latter here.

As shown in Table 8, LLM-based promtable methods, like LL3DA (Chen et al., 2024) exhibit high latency that is inadequate for autonomous driving deployments. In contrast, our TTC system introduces only a little extra latency compared to its base detector, which meets the real-time inference requirements and thus can be applicable for online autonomous driving systems.

## C.7 COMPARISON WITH OFFLINE FINE-TUNING USING USER FEEDBACK

In this section, we compare our TTC method with the approach that collects missing objects during inference and subsequently fine-tunes with the collected test-time 2D ground truth. This is another approach for utilizing test-time human feedback, though with delays in further model fine-tuning.

We select MV2D (Wang et al., 2023c) as the baseline since it relies on 2D detection results for 3D object detection, thus allowing it to utilize 2D human

Table 9: Comparisons between the TTC-MV2D and MV2D fine-tuned with feedback 2D annotations. "N=0" means having 2D feedbacks at every frame; "N=2" means less 2D feedbacks collected every 2 frames. TTC system, though without any extra training, still outperforms the offline fine-tuned MV2D, especially when annotations are limited (N=10).

| Exp. on Human Feedback collected with different frame interval $N$. | N | | | | | |
|---|---|---|---|---|---|---|
| | 0 | 2 | 4 | 6 | 8 | 10 |
| MV2D + Offline fine-tune | 44.6 | 42.7 | 41.4 | 40.7 | 39.8 | 39.0 |
| TTC-MV2D | 50.7 | 50.5 | 50.5 | 50.4 | 50.1 | 50.0 |

feedback annotations to fine-tune. Additionally, we collect 2D feedback from various frame intervals, as users cannot provide feedback at every frame. For MV2D, we use all the 2D box annotations

collected at specified intervals for fine-tuning. For the TTC-MV2D, we update the prompt buffer at these specified intervals.

As shown in Table 9, TTC-MV2D outperforms the MV2D model fine-tuned with 2D feedback. Notably, the performance of MV2D fine-tuned with human feedback from larger frame intervals, which means less frequent human feedback, declines significantly compared to models fine-tuned with feedback from every frame. In contrast, the TTC system allows the MV2D to perform effectively even when prompts are sourced from frames that are substantially different from the target frame. This finding highlights the practicality of the TTC design in utilizing a single corresponding visual prompt for streaming data, as it is impractical for users to provide 2D prompts at each frame.

## D QUALITATIVE RESULTS

We provide extensive visualizations to demonstrate the versatility of the TTC system across diverse scenarios:

- In Appendix D.1 and Appendix D.2, we fix the prompt buffer with visual prompts of either labeled or novel, unlabeled 3D objects from the nuScenes dataset to detect targets in the nuScenes images.

- In Appendix D.3, we test the performance with prompt buffer containing visual prompts in styles differing from the training distribution, such as Lego.

- In Appendix D.4, we visualize the similarity maps on out-of-domain images, including YouTube driving videos and Internet-sourced visual prompts, demonstrating the generalization of the visual prompt alignment and the effectiveness of our TTC in reducing driving risks in non-standard scenarios.

For all visualizations, the fixed prompt buffer is shown in the first row. These comprehensive evaluations highlight the versatility of the TTC system in leveraging diverse visual prompts for 3D detection. All results are conducted with TTC-MonoDETR.

### D.1 IN-DOMAIN VISUAL PROMPTS ON NUSCENES "SEEN" OBJECTS

This visualization focuses on the in-domain detection performance of the TTC system on the nuScenes dataset. We utilize visual prompts from *labeled* objects in the nuScenes dataset, and demonstrate the system's ability to effectively detect and track these target objects across different frames, as shown in Figure 13 and Figure 14. The results illustrate that our TTC can accurately localize and consistently track the target objects of interest within the nuScenes scenarios, showcasing its effectiveness in handling in-domain visual prompts.

### D.2 IN-DOMAIN VISUAL PROMPTS ON NUSCENES "UNSEEN" OBJECTS

This visualization focuses on the TTC's ability to handle visual prompts of objects *not labeled* in the nuScenes dataset. As shown in Figure 15, Figure 16, and Figure 17, our method demonstrates its potential to detect and track novel, out-of-distribution objects with these unseen visual prompts.

The results illustrate the TTC system's capability to go beyond the training distribution and effectively localize and track objects that were not part of the original labeled dataset. This showcases the versatility and generalization ability of the visual prompt-based framework, enabling the detection of previously unseen objects. These findings highlight the potential of the TTC system to continuously expand its object detection capabilities by incorporating user-provided visual prompts, even for objects that were not included in the initial training data.

### D.3 OUT-DOMAIN VISUAL PROMPTS ON NUSCENES "SEEN" OBJECTS

This visualization focuses on the TTC's performance with visual prompts in *styles different from the training distribution*. As shown in Figure 18 and Figure 19, the model can effectively detect target objects using visual prompts in various views and styles that diverge from the original training data.

These results demonstrate the potential of the TTC system to be extended to customized online 3D detection scenarios, where users can provide arbitrary visual prompts to guide the detection of objects of interest. The model's ability to handle prompts across diverse style domains highlights its flexibility and versatility, a key advantage for enabling user-centric, interactive 3D perception systems.

### D.4 OUT-DOMAIN VISUAL PROMPTS ON REAL-WORLD EXAMPLES

This visualization examines the generalization and robustness of TTC detectors in aligning visual prompts with the corresponding target objects in the input video stream. We select challenging driving scenarios involving unexpected animals running into the path of the ego vehicle. This is aimed at demonstrating the TTC system's capability in reducing online driving risks in such non-standard situations.

As shown in Figure 20 and Figure 21, our method can effectively localize non-expected animals with higher responses in the regions where animals located, even though it was trained solely on the nuScenes dataset. Figure 22, Figure 23, Figure 24, Figure 25 further present examples of detection in a real-world scenario containing both vehicles and animals. Our method can detect all objects with their visual prompts simultaneously.

This further exemplifies the strong generalization capability of our TTC system, underscoring its potential for effectively handling challenging, edge-case scenarios on roads. The ability to accurately detect and localize unexpected objects beyond the training distribution highlights the robustness of the proposed approach, a key requirement for reliable autonomous driving systems.

## E    LICENSE OF ASSETS

The adopted nuScenes dataset (Caesar et al., 2020) is distributed under a CC BY-NC-SA 4.0 license. We implement the model based on mmDet3D codebase (Contributors, 2020), which is released under the Apache 2.0 license.

We will publicly share our code and models upon acceptance under Apache License 2.0.

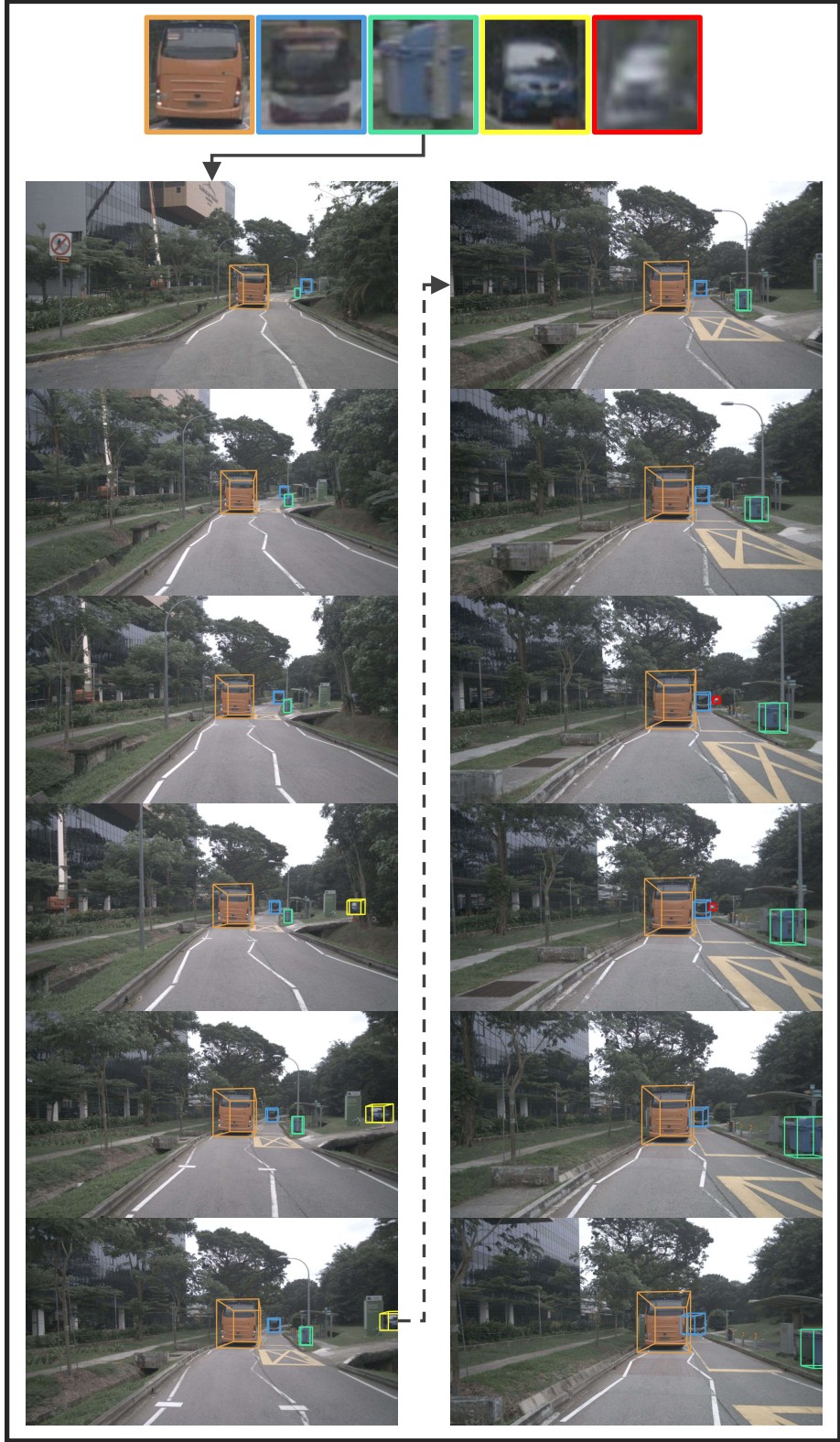

Figure 13: **Visualizations on nuScenes scenarios with in-domain visual prompts of** *labeled* **objects.** TTC system enables continuous 3D detection and tracking based on visual prompts. The images in the first row indicate the visual prompts in prompt buffer, and images in other rows represent 3D detection results prompted by the corresponding visual prompts. Different identities are indicated with different colors.

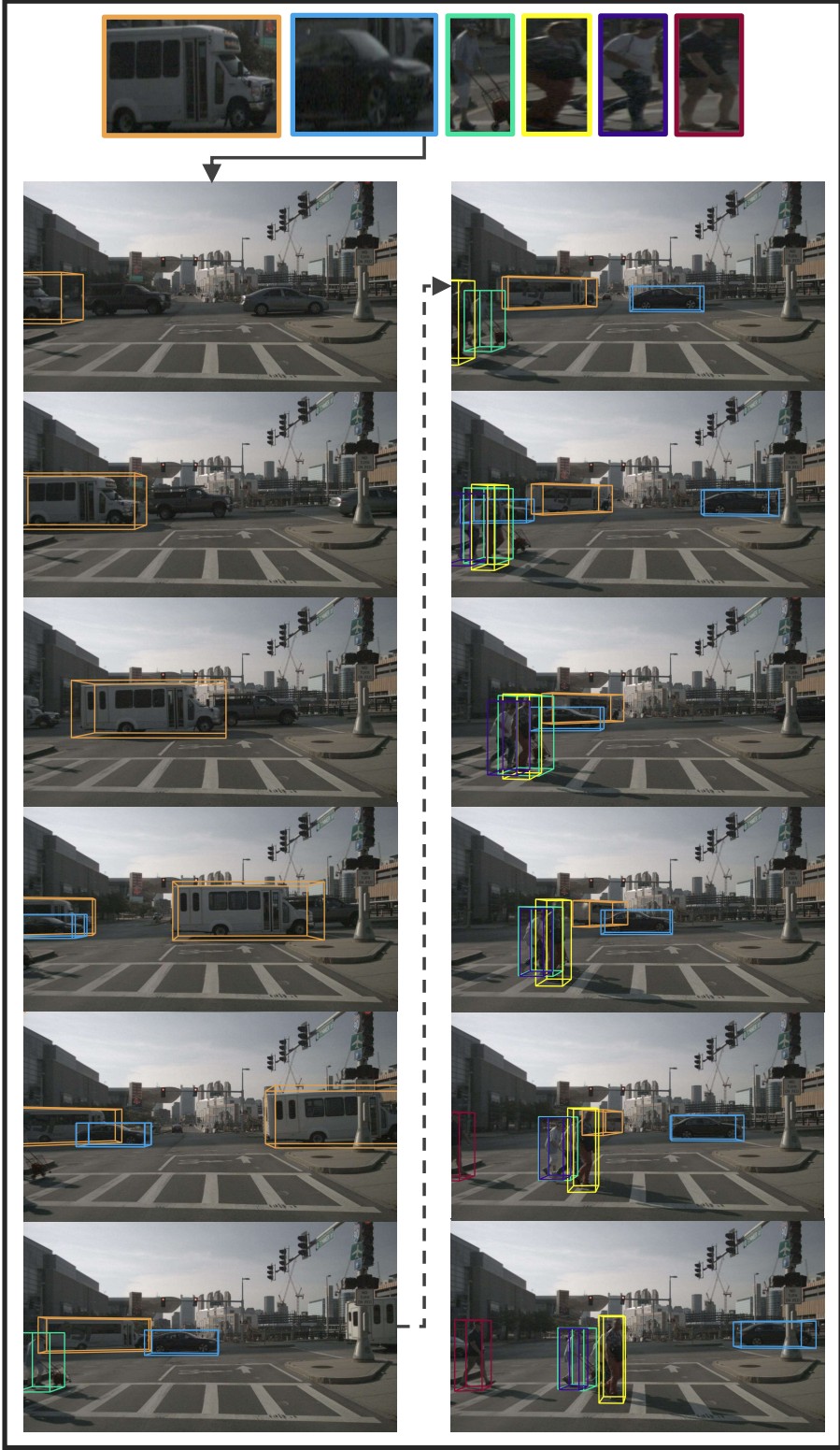

Figure 14: **Visualizations on nuScenes scenarios with in-domain visual prompts of** *labeled* **objects.** TTC system enables continuous 3D detection and tracking based on visual prompts. The images in the first row indicate the visual prompts in prompt buffer, and images in other rows represent 3D detection results prompted by the corresponding visual prompts. Different identities are indicated with different colors.

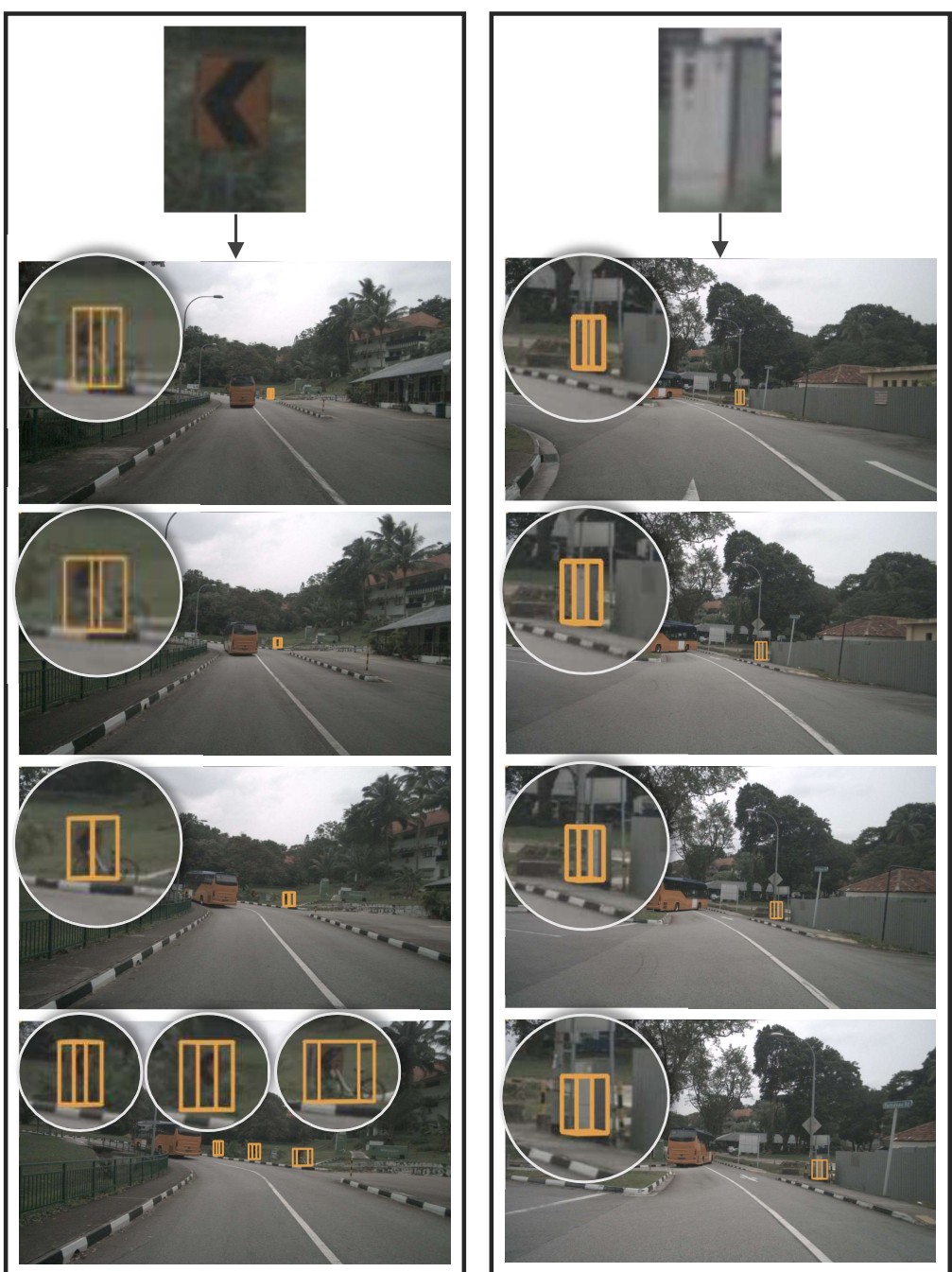

Figure 15: **Visualizations on nuScenes scenarios with in-domain visual prompts of *un-labeled* objects.** TTC system enables 3D detection and tracking of "novel" objects unseen during training. The image in the first row indicates the visual prompt in the prompt buffer, and images in other rows represent 3D detection results prompted by the corresponding visual prompts. Interestingly, with the one-to-N mapping mechanism of the visual prompt alignment, TTC system can detect multiple objects with similar visual descriptions to the visual prompt simultaneously.

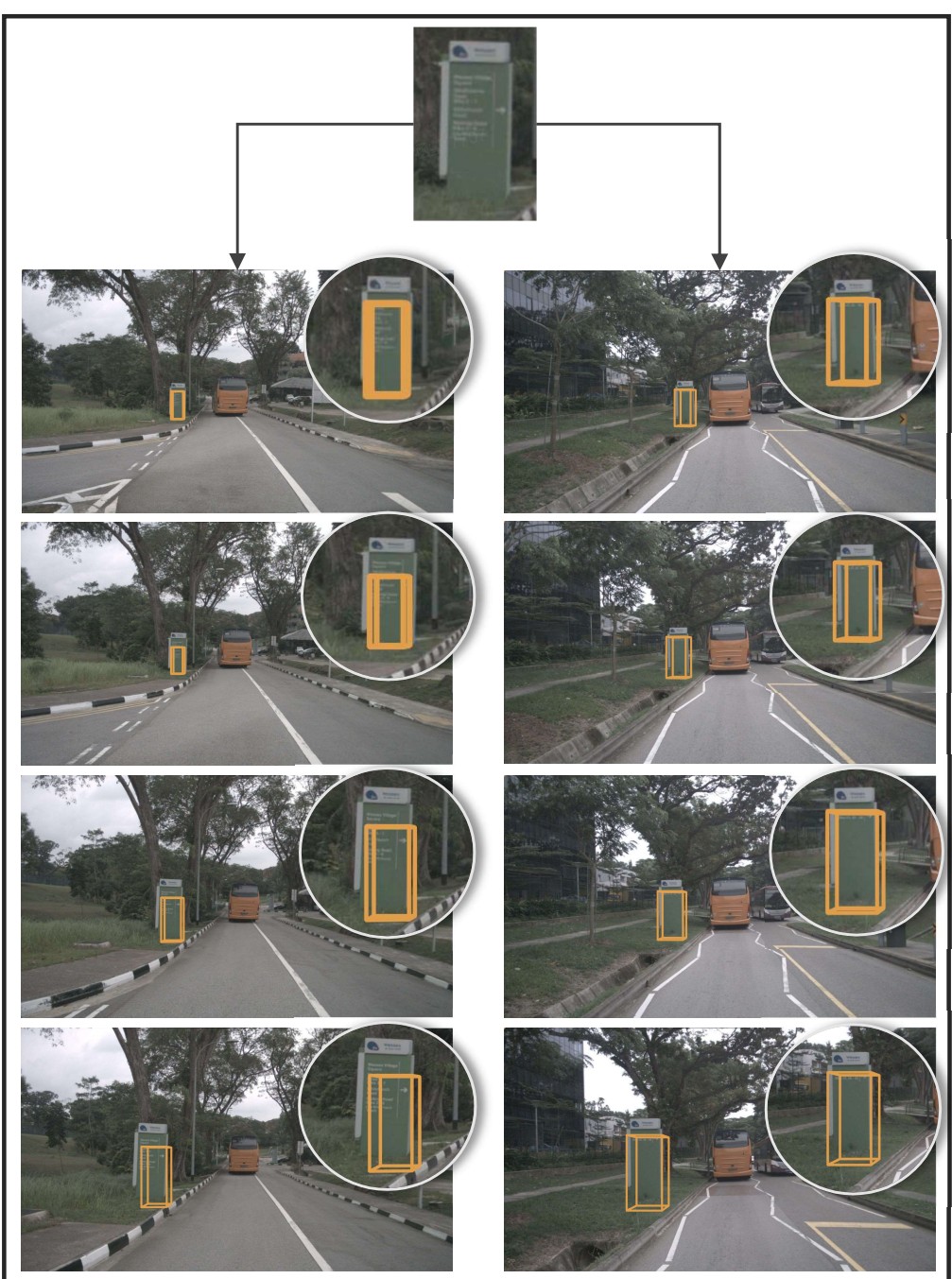

Figure 16: **Visualizations on nuScenes scenarios with in-domain visual prompts of *un-labeled objects.*** TTC system enables 3D detection and tracking of "novel" objects unseen during training. The image in the first row indicates the visual prompt in the prompt buffer, and images in other rows represent 3D detection results prompted by the corresponding visual prompts.

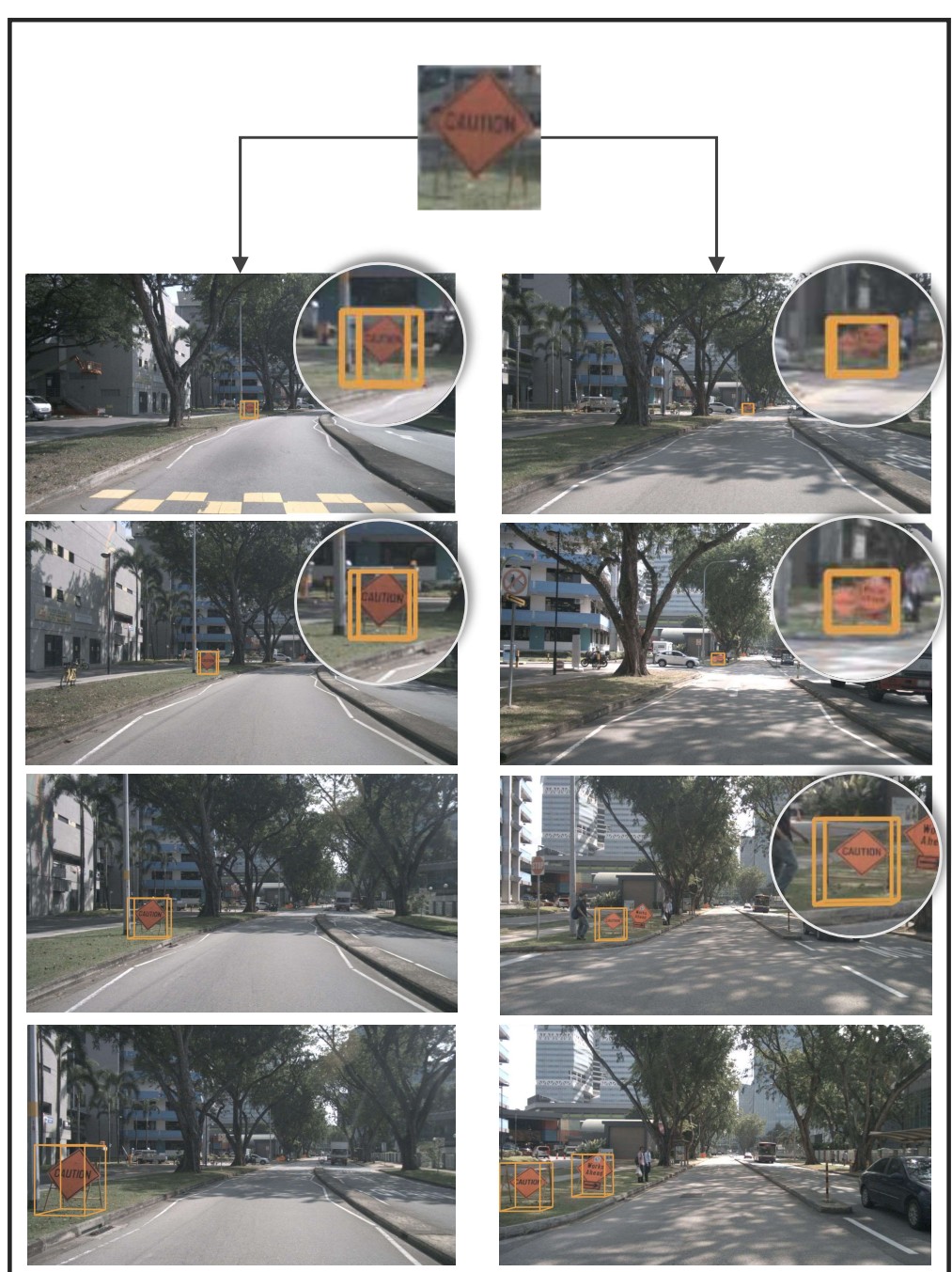

Figure 17: **Visualizations on nuScenes scenarios with in-domain visual prompts of *un-labeled objects*.** TTC system enables 3D detection and tracking of "novel" objects unseen during training. The image in the first row indicates the visual prompt in the prompt buffer, and images in other rows represent 3D detection results prompted by the corresponding visual prompts. Interestingly, with the one-to-N mapping mechanism of the visual prompt alignment, TTC system can detect multiple objects with similar visual descriptions to the visual prompt simultaneously.

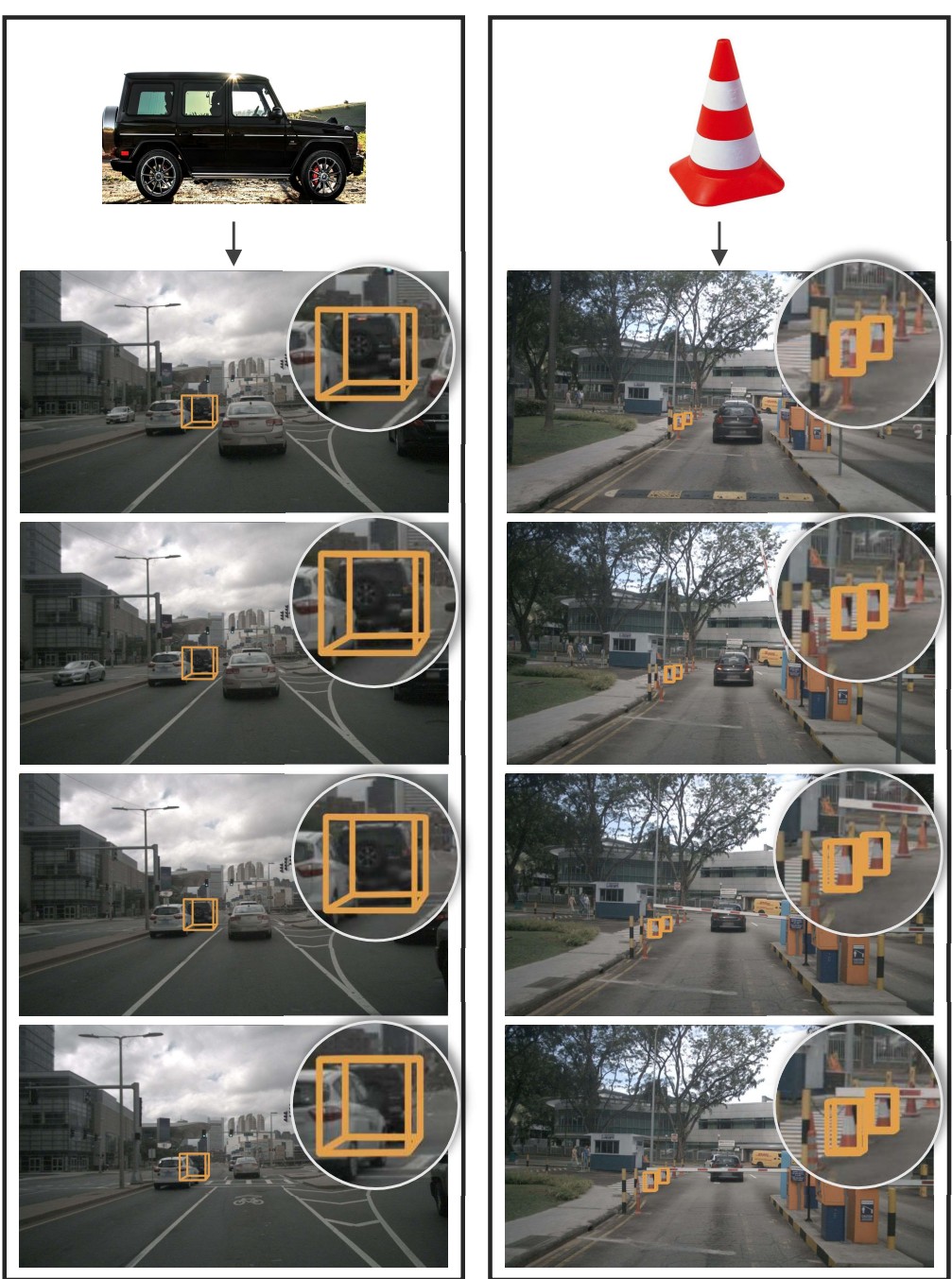

Figure 18: **Visualizations on nuScenes scenarios with *out-domain* visual prompts of *labeled* objects.** TTC system can perform 3D detection and tracking via visual prompts with arbitrary styles (**imagery style**). The image in the first row indicates the visual prompt in the prompt buffer, and images in other rows represent 3D detection results prompted by the corresponding visual prompts. With the one-to-N mapping mechanism of the visual prompt alignment, TTC detectors can detect multiple objects with similar visual descriptions to the visual prompt at the same time.

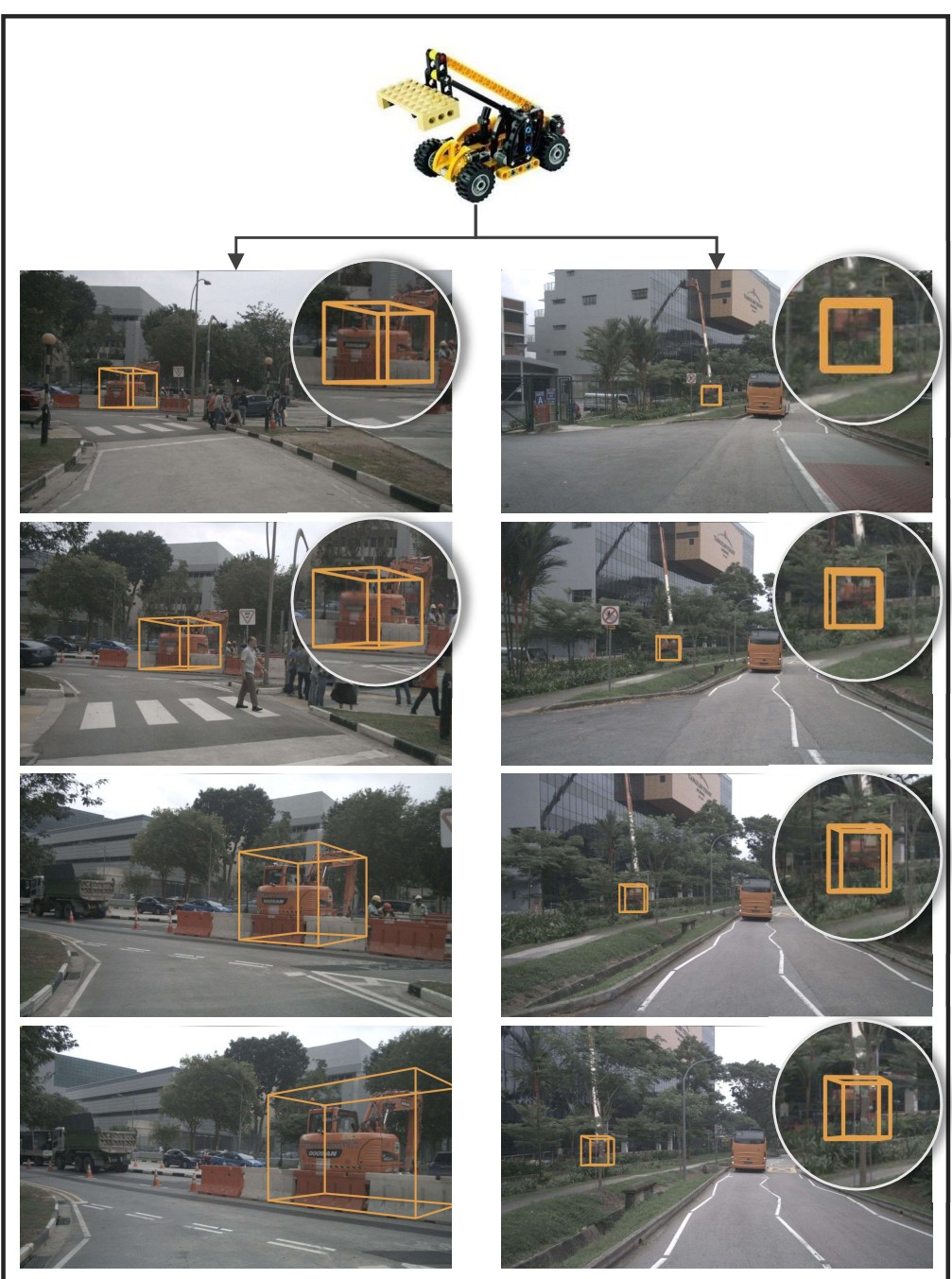

Figure 19: **Visualizations on nuScenes scenarios with *out-domain* visual prompts of *labeled* objects.** TTC system can perform 3D detection and tracking via visual prompts with arbitrary styles (**Lego style**). The image in the first row indicates the visual prompt in the prompt buffer, and images in other rows represent 3D detection results prompted by the corresponding visual prompts. With the one-to-N mapping mechanism of the visual prompt alignment, TTC detectors can detect multiple objects with similar visual descriptions to the visual prompt at the same time.

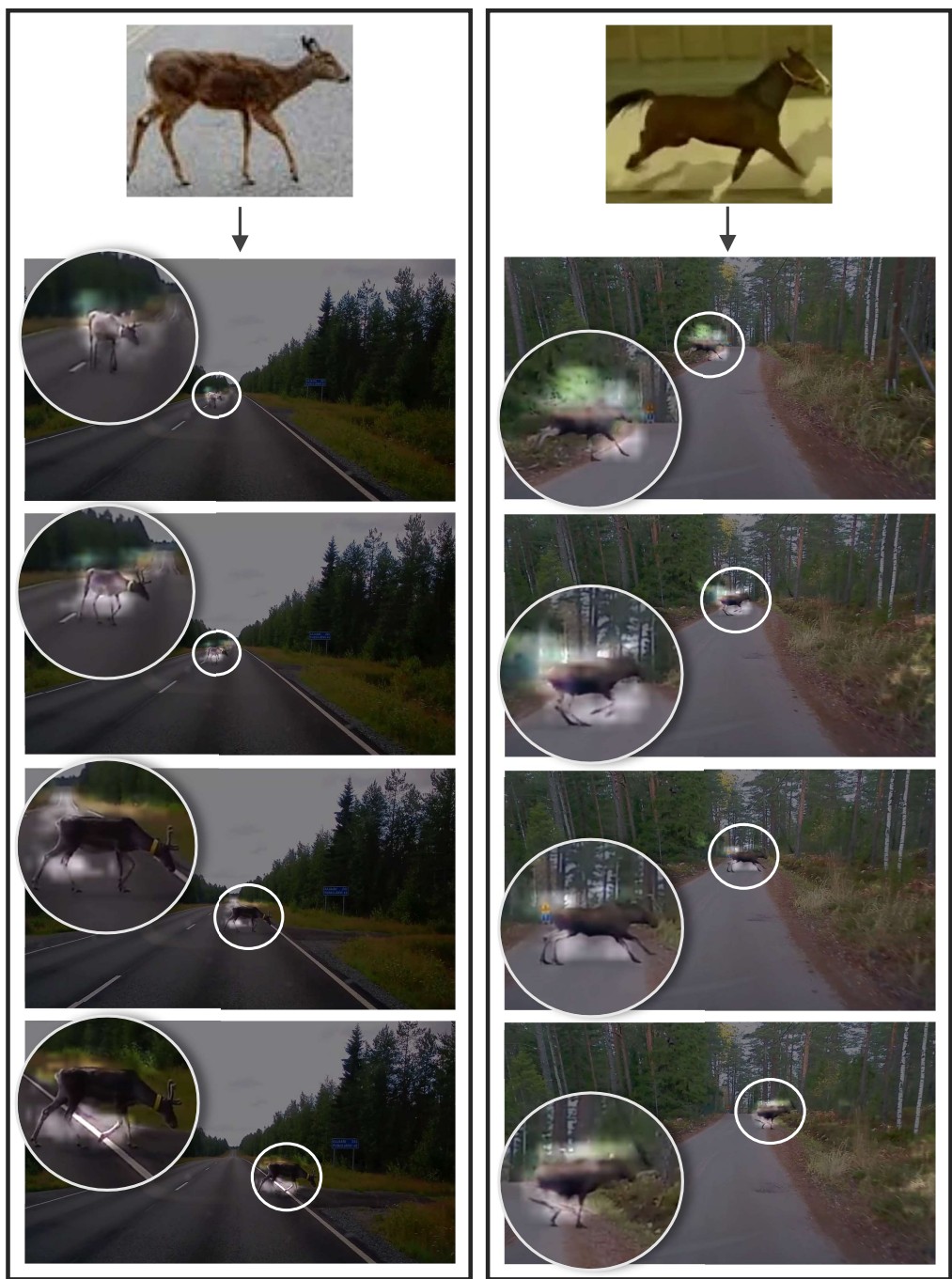

Figure 20: **Visualizations on real world scenarios with out-of-domain visual prompts.** At here, we show the similarity map from visual prompt alignment with visual prompts of arbitrary objects unseen during training. Brighter colors highlight higher responses. As illustrated, our method works well in novel scenarios with visual prompts of arbitrary object-of-interests. Best viewed in color.

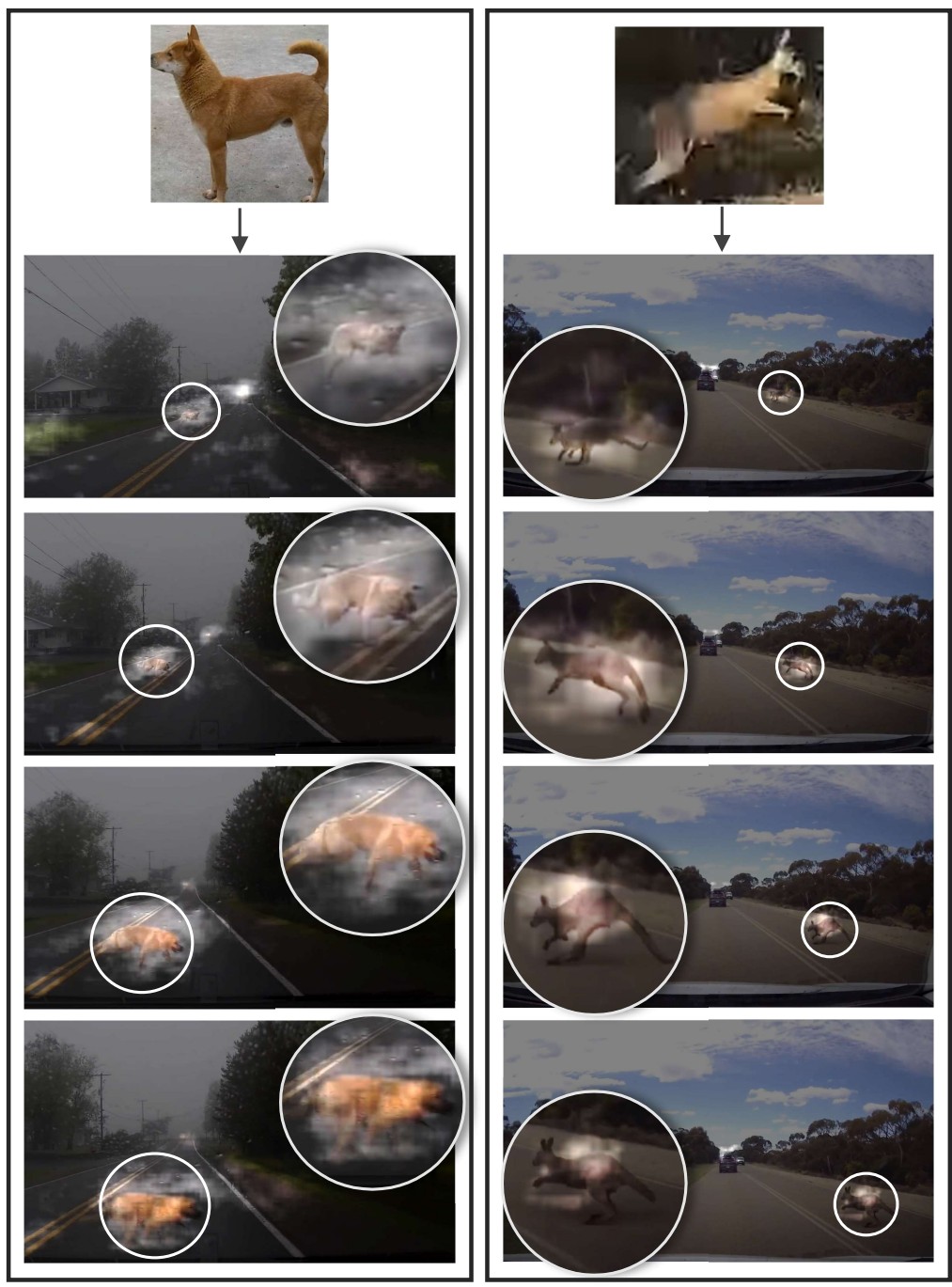

Figure 21: **Visualizations on real world scenarios with out-of-domain visual prompts.** At here, we show the similarity map from visual prompt alignment with visual prompts of arbitrary objects unseen during training. Brighter colors highlight higher responses. As illustrated, our method works well in novel scenarios with visual prompts of arbitrary object-of-interests. Best viewed in color.

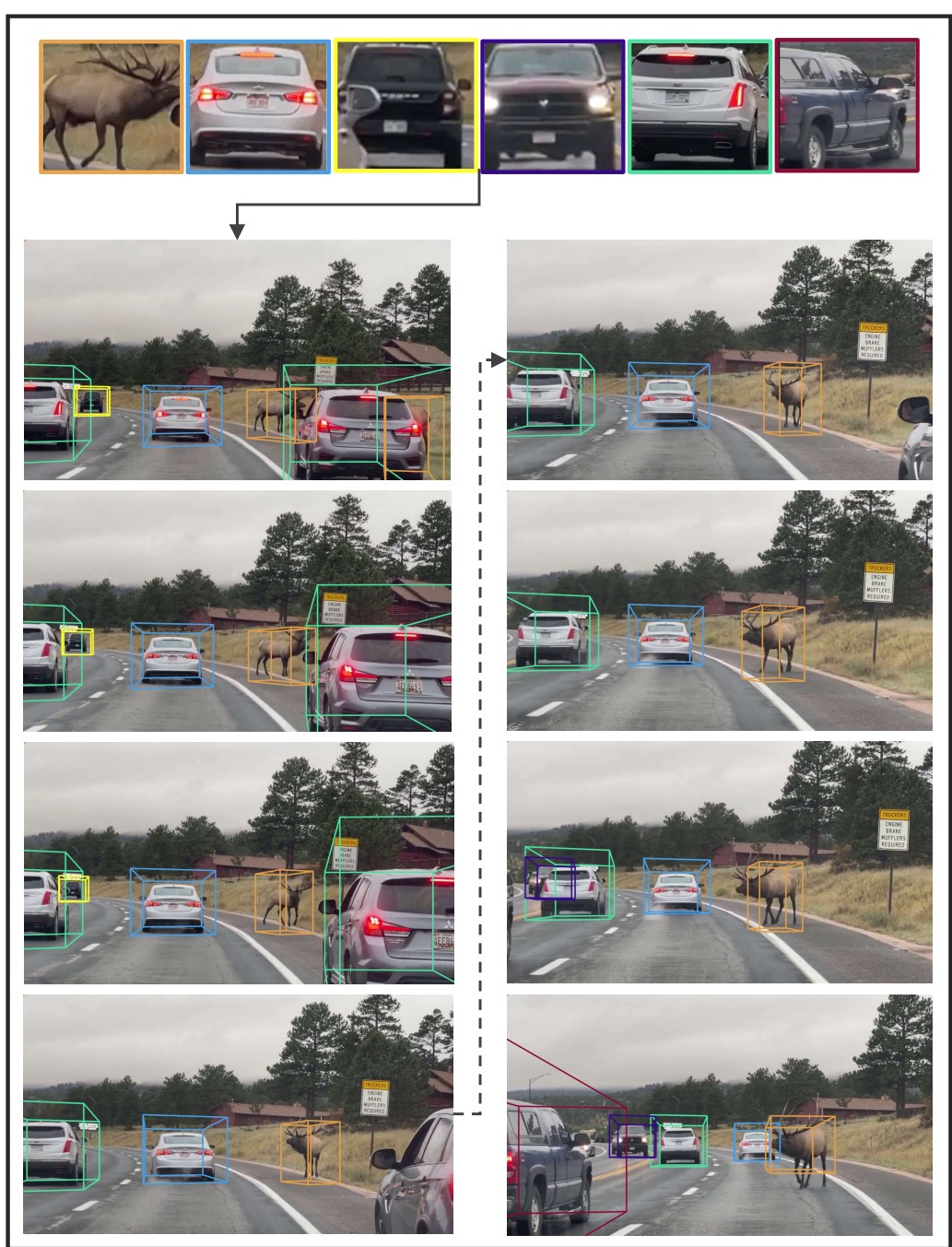

Figure 22: **Visualizations on real world scenarios with out-of-domain visual prompts.** This figure demonstrates the TTC's effectiveness in real-world 3D detection, even with visual prompts of unseen objects. demonstrate a case of real-world detection As illustrated, our method works well in real-world scenarios with visual prompts of unseen objects. Different object identities are indicated by distinct colors.

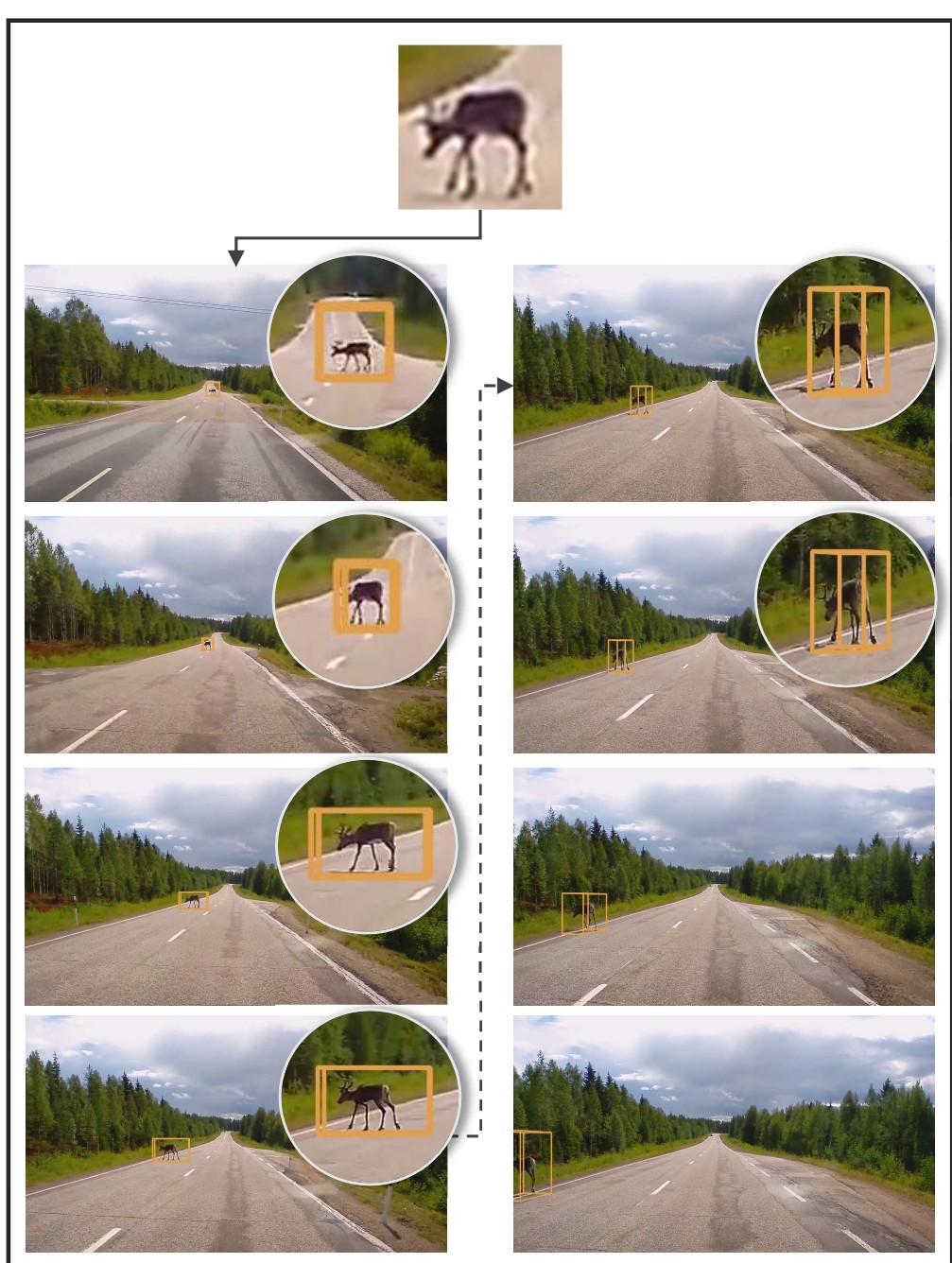

Figure 23: **Visualizations on real world scenarios with out-of-domain visual prompts.** This figure demonstrates a case of real-world 3D object detection. We provide the visual prompt from the same scene as the target object, though it is unseen during training, and our TTC system then detects the target objects in the subsequent video frames.

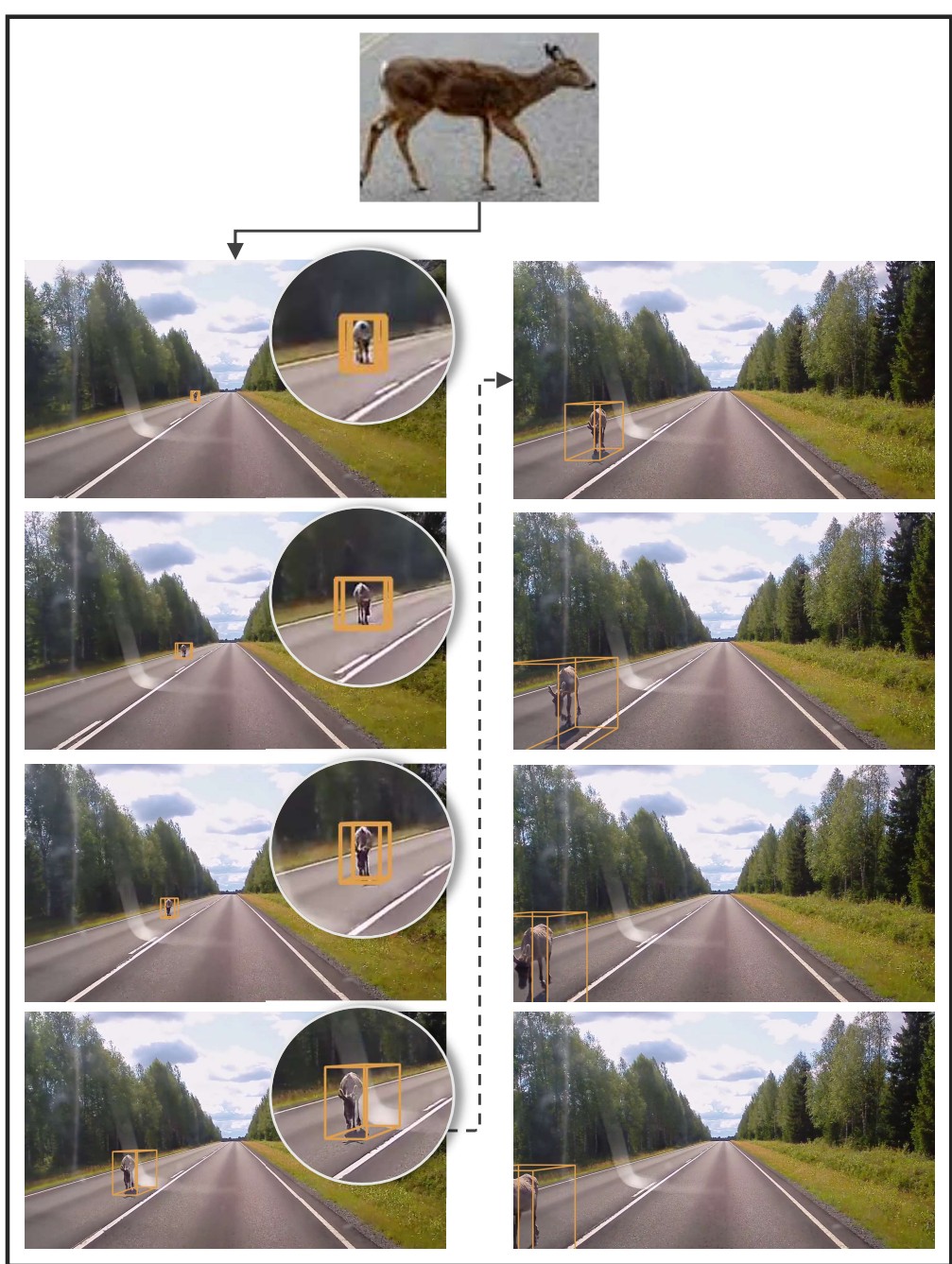

Figure 24: **Visualizations on real world scenarios with out-of-domain visual prompts.** This figure demonstrates a case of real-world 3D object detection. We provide a visual prompt from a separate image as the pre-defined prompt and visualize it at the beginning of the sequence. As described, our TTC system can successfully detect the "Deer" object using this stylized deer prompt sourced from the internet. This example highlights the TTC's capability to effectively leverage diverse, user-supplied visual prompts to accurately identify target objects, even when the prompts are not directly from the same scene.

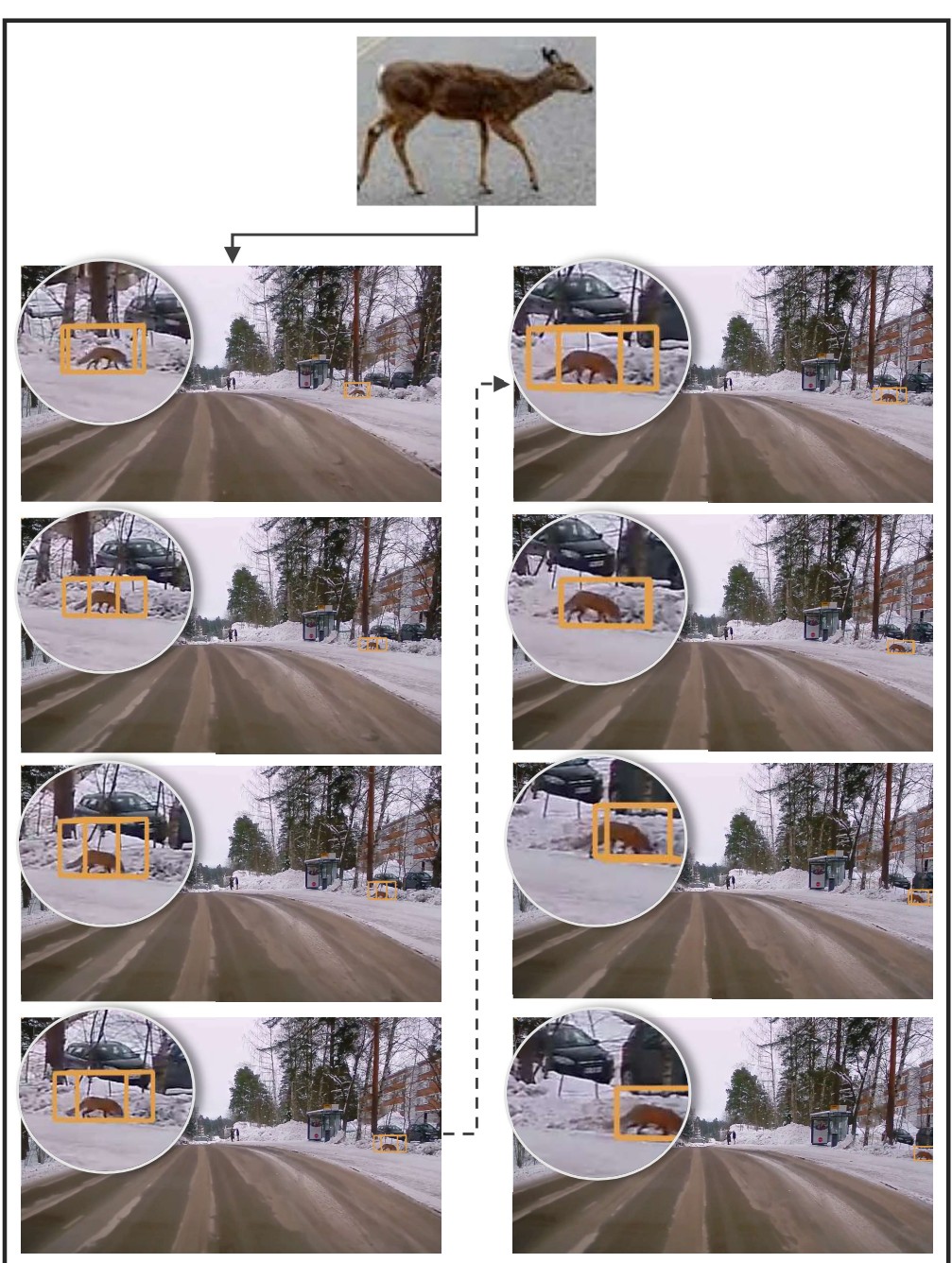

Figure 25: **Visualizations on real world scenarios with out-of-domain visual prompts.** Another case demonstrates the use of an internet-sourced "Deer" visual prompt to detect the deer in a different real-world scenario. As shown, our TTC system effectively detects the deer even when it is partially obscured by snow. This example further illustrates the robust performance of the TTC framework in accurately localizing target objects, even in challenging environmental conditions, by leveraging flexible visual prompts provided by users.

