# OpenReview forum: "Test-time Correction with Human Feedback: An Online 3D Detection System via Visual Prompting"
_ICLR.cc/2025/Conference — ICLR 2025 Conference Withdrawn Submission_

### Official Review · Reviewer_R52R · 2024-11-03

**Soundness:** 2
**Presentation:** 3
**Contribution:** 2
**Rating:** 5
**Confidence:** 5

**Summary:**

This paper introduces Test-time Correction (TTC) system, a online 3D detection system designated for online correction of test-time errors via human feedback. The proposed TTC system includes two components: Online Adapter (OA) that enables 3D detectors with visual promotable ability, and a visual prompt buffer that records missing objects. Experiments were conducted on the nuScenes dataset, focusing on the TTC system across various 3D detectors and in out-of-training-distribution scenarios.

**Strengths:**

1. This paper is clearly structured and written-well.
2. This paper focuses on an interesting issue, namely that a online 3D detection system designated for online correction of test-time errors via human feedback.

**Weaknesses:**

1. The rationale behind this task setup requires further discussion. Since the proposed task involves human feedback, this process is often uncontrollable, making it challenging to ensure real-time performance. This limitation affects the feasibility of applying the task in real-world autonomous driving scenarios.
2. The rationale behind the EDS evaluation metric requires further discussion. Classification accuracy is also crucial for autonomous driving, and focusing solely on localization performance while neglecting classification performance is not realistic.
3. The proposed method is only applicable to vision-based autonomous driving solutions, limiting its generalizability to LiDAR-based autonomous driving systems.

**Questions:**

1. Please refer to the Paper Weaknesses mentioned above.
2. The experimental section only conducts multiple tasks on MonoDETR; however, multi-view 3D detection is currently the mainstream approach in autonomous driving solutions. It is recommended to include experiments on mainstream multi-view 3D detectors across various tasks.
3. The proposed method focuses solely on correcting missed detections, yet false positives are also a significant issue in autonomous driving. Is there scalability for correcting false positives?
4. The experimental section lacks comparisons with existing instruction-based 3D detection methods typically utilize text, boxes, or clicks as prompts; it is recommended to include such comparisons.

---

### Official Review · Reviewer_2P2Q · 2024-11-04

**Soundness:** 2
**Presentation:** 3
**Contribution:** 2
**Rating:** 5
**Confidence:** 3

**Summary:**

This paper presents the Test-time Correction (TTC) system, an online 3D detection framework designed to correct test-time errors through real-time human feedback. Unlike conventional offline static 3D detectors, TTC aims to learn real-time error rectification by incorporating user feedback (e.g., clicks or bounding boxes). This approach allows for immediate updates to detection results for subsequent streaming inputs, even when the model operates with fixed parameters. The TTC system is achieved by integrating an OA module, an online adapter with a prompt-driven architecture, into existing 3D detectors for real-time correction. The key is visual prompts, specifically images of missed objects, which guide both current detection and future tracking. These visual prompts, representing objects missed during inference, are stored in a buffer to support continuous error correction across frames. Extensive experiments reveal substantial improvements in immediate error rectification compared to pre-trained 3D detectors, even under limited labeling, zero-shot detection, and challenging environmental conditions.

**Strengths:**

1. This paper is well written, and easy to understand.

2. Enhancing 3D detection is important task for autonomous driving.

3. The performance gain is impressive.

**Weaknesses:**

1. Motivation

Pretrained static 3D detection modules have clear limitations due to issues like domain gaps, and I agree with the goal of improving these. However, I find it difficult to fully empathize with the motivation behind TTC. If issues in the predictions of the 3D detection module indeed pose a significant impact on safety, as stated, would there realistically be an opportunity to perform online correction in such scenarios?

Additionally, I am skeptical about the feasibility of interventions like visual prompting during driving. Operating devices such as navigation systems manually while driving is likely a legal violation in most countries, and in practice, the difficulty level for performing such tasks during driving seems exceedingly high.

2. Comparison with TTA or TTT

In this field, there are various approaches for online improvement of pre-trained static models, such as test-time adaptation (TTA) and test-time training (TTT). Notably, most of these methods function without the need for human feedback. A thorough methodological and performance comparison with these approaches is essential. Additionally, while TTT may be somewhat challenging, in the case of TTA, it seems feasible to utilize human feedback as a direct learning guidance. I would appreciate a more in-depth discussion on this aspect as well.

3. Robustness

It is unrealistic to expect that user corrections will always be accurate. Depending on the situation, incorrect user interventions could potentially worsen the proposed TTC. It would be beneficial to model the noise that might exist in visual prompting and demonstrate that TTC can operate robustly even when this noise becomes more pronounced.

**Questions:**

Please refer to the weaknesses.

---

### Official Review · Reviewer_d1Gg · 2024-11-04

**Soundness:** 3
**Presentation:** 2
**Contribution:** 2
**Rating:** 3
**Confidence:** 4

**Summary:**

This paper presents the Test-time Correction (TTC) system, a novel online 3D detection framework designed to correct test-time errors through human feedback. This approach aims to enhance the safety of deployed autonomous driving systems.

The key idea of the TTC system is to improve existing 3D detectors with the Online Adapter (OA) module, a prompt-driven design that facilitates real-time error correction. Central to the OA module are visual prompts—images of missed objects of interest that guide corresponding detection and subsequent tracking. These visual prompts are stored in a visual prompt buffer to enable continuous error correction in subsequent frames. This approach allows the TTC system to consistently detect missed objects in real-time, thereby effectively reducing driving risks.

Experimental results show that the proposed method, through test-time rectification, enhances the performance of offline monocular detectors (Zhang et al., 2022a), multi-view detectors (Wang et al., 2023c), and BEV detectors (Yang et al., 2023a) without the need for additional training.

**Strengths:**

The broader impact of this paper may lie in inspiring the research community to further investigate the online rectification approach in autonomous driving systems. This crucial technology has the potential to significantly enhance the safety and reliability of safety-critical applications.

**Weaknesses:**

The paper lacks self-containment. For example, in lines 217-231, where the authors describe various visual prompts, they mainly reference numerous other methods without offering sufficient detail. This heavy reliance on external sources renders the paper somewhat incremental, as it fails to clearly articulate the novel contributions and context of the visual prompts within the proposed framework. Furthermore, this lack of clarity results in the use of many notations, such as "visual features" and "image features," without providing clear definitions.


Rather than referring to it as "visual prompts," the pipeline developed in this paper essentially provides a template containing location and size information in the buffer, enabling generic tracking during test time without any additional training. Therefore, the authors are encouraged to clarify whether this pipeline fundamentally differs from a single-object tracker. Additionally, it would be beneficial to include an experiment comparing state-of-the-art (SOTA) trackers for test-time correction as part of the evaluation

**Questions:**

"To achieve such TTC system, we equip existing 3D detectors with OA module, an online adapter with prompt-driven design for online correction." However, the acronym "OA" is not defined in the abstract.

---

### Official Review · Reviewer_Mvta · 2024-11-04

**Soundness:** 3
**Presentation:** 3
**Contribution:** 3
**Rating:** 6
**Confidence:** 3

**Summary:**

This paper introduces a Test-time Correction (TTC) method that leverages human feedback to correct errors in real-time during testing. The core component, the Online Adapter (OA) module, enables existing 3D detectors to use visual prompts for continuously detecting previously undetected 3D objects.

**Strengths:**

1. Clear and well-structured writing, and easy to understand.
2. Significant performance improvements across multiple 3D detectors and comprehensive ablation studies validate module effectiveness.
3. The OA module is simple but effective.

**Weaknesses:**

1. Re-entering the regions of undetected targets as visual prompts introduces human knowledge, which may lead to potential biases and affect the fairness of comparative experiments.
2. The TTC method needs to maintain a buffer of visual cues and solve the matching problem between cues and target objects, which increases the complexity.
3. The experimental section lacks a description of how the visual prompts used during testing are obtained.

**Questions:**

1. How are the visual prompts used during testing obtained？I'm not sure if it's just adding the corresponding areas of undetected targets to the visual cue buffer.
2. Could the TTC method be combined with LLM-based 3D detection approaches to enhance generalization for novel object categories and domain shifts?
3. How does the TTC method handle potential noise and latency in user feedback?

---

### Official Review · Reviewer_cfWC · 2024-11-04

**Soundness:** 2
**Presentation:** 2
**Contribution:** 1
**Rating:** 3
**Confidence:** 4

**Summary:**

This paper presents a 3D object adaptation method that could adapt the human feedback online. The system could work with various visual prompts including reference images, box in the image and click in the image. The proposed methods is validated on both in domain and out of domain dataset, and demonstrating its effectiveness in these scenarios.

**Strengths:**

1. The designed validation experiments are comprehensive.
2. The proposed method is training-free, which makes it can be broadly applied.

**Weaknesses:**

1. The proposed can only handle missing objects. It seems it cannot reduce the FP in test time.
2. Although the authors explain the differences between proposed TTC and single object tracking, the explanation is unconvincing. The visual and object prompt can be easily used in SOT setting. Such discussion and at least comparisons with bbox annotations are must. This is the key concern in my evaluation.

**Questions:**

1. The paper does not clearly present how the proposed modules are trained, especially for the two layers MLP in key online adapter module.

---

### Official Review · Reviewer_ECUi · 2024-11-08

**Soundness:** 3
**Presentation:** 3
**Contribution:** 3
**Rating:** 6
**Confidence:** 3

**Summary:**

The authors introduce the Test-time Correction (TTC) system, an innovative online 3D detection framework designed for correcting test-time errors in real-time through human feedback. TTC demonstrates the capability for immediate error rectification. Extensive experiments show substantial improvements in real-time error correction over pre-trained 3D detectors, even in challenging scenarios involving limited labels, zero-shot detection, and adverse conditions.

**Strengths:**

1. Model flexibility: Accepts both monocular and multi-view data and supports any combination of various prompts (object, box, point, and novel visual prompts).
2. Clarity of writing: The paper is well-written, logically structured, and easy to read.
3. Extensive experiments: The main text and supplementary materials provide ample experiments to validate the effectiveness of TTC.
Practical feasibility: The authors explain real-world application scenarios, achieving immediate error rectification through user-friendly prompts.

**Weaknesses:**

1. OA is one of the core modules of the model, serving as a bridge between prompts and offline-trained 3D detectors. However, the explanation of OA in the method section is somewhat abstract; adding simple illustrative diagrams could aid understanding.
2. In the Related Work section, the Online 3D Detection System subsection discusses online 3D detectors. Expanding on offline 3D detectors would help readers better understand the development of offline versus online 3D detection.
3. There are some minor typos in the text that need correction.

**Questions:**

It is recommended to include the full term "Online Adapter (OA)" the first time OA is mentioned in the abstract.

---

### Note · Authors · 2024-11-14

I have read and agree with the venue's withdrawal policy on behalf of myself and my co-authors.